# Spatial Reasoning with Denoising Models

**Christopher Wewer** [1]   **Bart Pogodzinski** [1]   **Bernt Schiele** [1]   **Jan Eric Lenssen** [1]

Playing MNIST Sudoku with Image Generators....

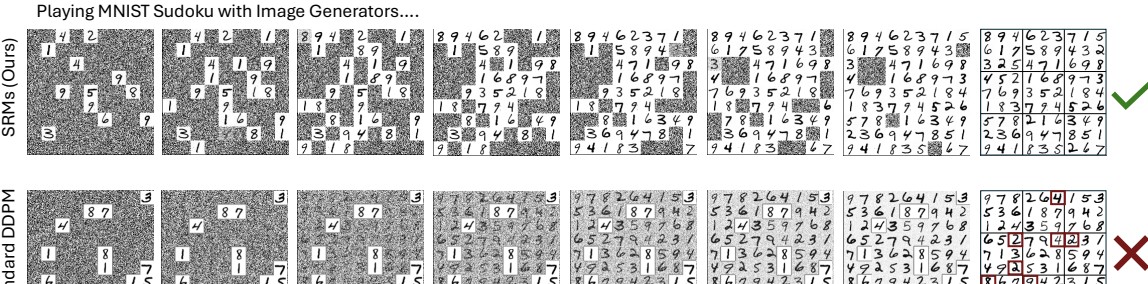

Figure 1: We introduce **Spatial Reasoning Models (SRMs)**, a framework to systematically investigate diffusion/flow-based generative models with respect to their reasoning capabilities over multiple variables. We introduce benchmarks that allow quantification of higher-level reasoning capabilities and show that manual and automatic schemes for sequentialization can heavily reduce hallucination. Here, we show the solving process of one of the benchmarks, a Sudoku game consisting of MNIST images, which is solved correctly by our SRMs, while standard diffusion models fail.

## Abstract

We introduce Spatial Reasoning Models (SRMs), a framework to perform reasoning over sets of continuous variables via denoising generative models. SRMs infer continuous representations on a set of unobserved variables, given observations on observed variables. Current generative models on spatial domains, such as diffusion and flow matching models, often collapse to hallucination in case of complex distributions. To measure this, we introduce a set of benchmark tasks that test the quality of complex reasoning in generative models and can quantify hallucination. The SRM framework allows to report key findings about importance of sequentialization in generation, the associated order, as well as the sampling strategies during training. It demonstrates, for the first time, that order of generation can successfully be predicted by the denoising network itself. Using these findings, we can increase the accuracy of specific reasoning tasks from $< 1\%$ to $> 50\%$. Our project website provides additional videos, code, and the benchmark datasets.

## 1. Introduction

Conditional generative models, such as GPTs (Radford et al., 2018) or denoising models like diffusion/flow-based models (Ho et al., 2020; Song et al., 2021; Lipman et al., 2023; Liu et al., 2023), promise significant advances in practical reasoning. They allow to model highly complex and multimodal distributions by learning to sample from them. While reasoning capabilities of LLMs are extensively explored in many recent works (Huang & Chang, 2023), similar efforts in continuous, spatial domains are needed to profit from the semantic structure that can be learned from high-dimensional, continuous data. Our work approaches this goal by providing a novel framework to benchmark and advance the reasoning capabilities of diffusion/flow-based generative models, which we deem of large importance for the further development of large image, video, and physically-grounded world models (Agarwal et al., 2025).

The process of reasoning can be defined as inferring the states of unobserved variables $x_i$, given a distinct set of observed variables $y_j$ (Kwan et al., 2008), with varying dependencies between all variables, loosely inspired by traditional fields of probabilistic graphical models and Bayesian networks. Most reasoning tasks are subject to a high degree of inherent uncertainty due to incompleteness of information given in the observed variables. Thus, it is natural to tackle reasoning problems with probabilistic inference, i.e., modeling $p(x_1, ..., x_n \mid y_1, ..., y_m)$. When reasoning across higher-dimensional continuous domains, such as images or image patches, these distributions become complex and multi-modal, preventing the application of traditional ap-

[1]Max Planck Institute for Informatics, Saarland Informatics Campus, Germany. Correspondence to: Christopher Wewer <cwewer@mpi-inf.mpg.de>, Bart Pogodzinski <bpogodzi@mpi-inf.mpg.de>, Jan Eric Lenssen <jlenssen@mpi-inf.mpg.de>.

*Proceedings of the 42nd International Conference on Machine Learning*, Vancouver, Canada. PMLR 267, 2025. Copyright 2025 by the author(s).

(a) Training Data                    (b) Conditioning image and generation output, standard DDPM

Figure 2: **MNIST Sudoku.** One of our reasoning benchmarks. (a) A dataset of correct Sudokus, consisting of random MNIST images. (b) When training a conditional diffusion model on correct examples it fails to conditionally generate correct solutions for hard Sudokus (almost 0% accuracy). In contrast, SRMs achieve $> 50\%$ accuracy for these cases.

proaches that rely on Gaussian assumptions or predefined families of discrete distributions.

Generative models for spatial domains typically learn to sample from an approximated data distribution $p(x_1, .., x_n)$. To perform reasoning, it is desired that such a model *abstracts* from the low-level data space and is able to detect high-level patterns that occur in seen data. To obtain an insight into the level of such capabilities of existing diffusion/flow models, we introduce a set of benchmarks. One example based on the well-known Sudoku game, where each number from 1 to 9 must occur exactly once in each row, column and 3x3 block, is shown in Fig. 2. A (conditional) diffusion model is trained to generate or complete correct Sudokus consisting of varying MNIST digits. While the generated images look realistic on the first glance, as individual numbers are high-quality samples from the MNIST dataset, the generation process fails to sample correct instances. We identify this effect as an instance of *hallucination*, where the model falls back to superficial solutions that satisfy the pixel-wise MSE loss well enough, in case it fails to capture the actual, complex patterns in the underlying distribution.

In the space of large language models (LLMs), *chain-of-thought* prompting has achieved successes in fighting such hallucinations by prompting the model to make smaller, sequential steps towards the solution (Wei et al., 2022). Inspired by these advances, we develop and investigate strategies in our spatial setting that can lead to similar effects. The chain-rule of probabilities lets us decompose

$$p(x_1, .., x_n) = \prod_{i=1}^{n} p(x_{\pi(i)} | \{x_{\pi(j)}\}_{j=i+1}^{n}) \qquad (1)$$

for arbitrary permutations $\pi$. While all orders $\pi$ model the correct distribution in theory, the individual orders lead to chains of varying complexity in the individual distributions, determined by hidden dependencies, potentially leading to varying amounts of hallucination during sampling. We believe investigating strategies to find (1) the correct amount of sequentialization and (2) the best order with least amount of hallucination is a promising venture for further development of continuous generative models.

To this end, we introduce *Spatial Reasoning Models (SRMs)*, an architecture-agnostic framework to formulate (soft or hard) sequentialization strategies that can be used for reasoning on sets of continuous variables without canonical order. Within this framework, we create multiple of such strategies and perform an exhaustive evaluation on benchmark tasks as described above.

In summary, our **contributions** are

- a general framework for reasoning over sets of continuous variables with denoising generative models,

- a novel algorithm for $t$-sampling during training, which adjusts for multiple variables,

- different task-specific and task-agnostic, soft and hard variants of sequentialization within reasoning models,

- a benchmark to quantitatively measure hallucination in reasoning over visual domains.

We can summarize the **key findings** as

- diffusion/flow-based generative models are capable of simple visual reasoning but fall back to *hallucination* when the modeled distribution becomes too complex,

- inducing sequentialization can simplify the task by decomposing into simpler distributions, reducing hallucination, and improving reasoning,

- greedy orders based on predicted uncertainty can significantly improve reasoning quality further,

- the choice of training $t$-sampling strategies is crucial when adjusting for sequentialization.

Our framework, code, and benchmarks are available on our project website for further investigation and development.

## 2. Related Work

Probabilistic reasoning is currently receiving a lot of attention, mostly driven by recent advancements in large language models (LLMs) (Huang & Chang, 2023; Plaat et al., 2024) that allow to generate tokens from a discrete codebook. To leverage their capabilities, many modalities

have been mapped to such discrete spaces, using specialized tokenizers (van den Oord et al., 2017), allowing to reason about originally continuous domains to certain degrees (Chen et al., 2023). Vision Language Models (VLMs) have been heavily used to connect spatial domains with language, to perform reasoning about images in the text domain (Chen et al., 2024b). In this work, we move in a different direction and investigate reasoning capabilities across continuous domains by generative models that have explicitly been designed to work on these domains, such as DDPM (Ho et al., 2020), DDIM (Song et al., 2021), rectified flow (Liu et al., 2023), or flow matching (Lipman et al., 2023). Investigating reasoning capabilities of such models has remained largely unexplored so far.

Closest to our work are recent methods that introduce differing levels of noise in sequential generation with diffusion models (Wu et al., 2023; Zhang et al., 2024; Ruhe et al., 2024; Chen et al., 2024a). AR-Diffusion (Wu et al., 2023) performed generation via diffusion on text spaces. Later, the idea was extended to sequential, continuous spaces, such as human poses (Zhang et al., 2024), or videos (Ruhe et al., 2024; Chen et al., 2024a), which also introduced overlapping via different noise levels of variables in purely sequential settings. Our work extends this direction to arbitrary spatial settings without canonical orders. Also, we heavily improve on the training sampling strategy used in Diffusion Forcing (Chen et al., 2024a). Another recent related work is MAR (Li et al., 2024), which performs autoregressive generation of image patches via denoising in random order.

## 3. Spatial Reasoning Models

In this section, we introduce our spatial reasoning models. We begin with formulating our general framework in Sec. 3.1. After training a SRM as outlined in Sec. 3.2, it can be leveraged with various sampling strategies. We introduce a selection of those in Sec. 3.3.

### 3.1. General Framework

Our goal is to learn to reason over sets of *continuous* random variables. Practical examples of such variables can be patches of images, frames of videos, or even multiple views of a 3D scene. For our SRM setting, we define *reasoning* over a set of continuous random variables as sampling

$$\hat{x}_1^{t_1}, ..., \hat{x}_n^{t_n} \sim q(x_1^{t_1}, ..., x_n^{t_n} \mid x_1^{t'_1}, ..., x_n^{t'_n}), \quad (2)$$

where $x_i^{t_i}$ denotes the variable $x_i$ with noise level $t_i$ during the denoising process and $t_i \leq t'_i$. Spatiality is (optionally) encoded via positional encodings on the variables. Choosing noise levels $t_i$ allows explicit control over amount and order of sequentialization, as discussed further in Sec. 3.1.2.

**Belief Propagation** The above paradigm is loosely inspired by belief propagation through Probabilistic Graphical Models (Pearl, 1982), propagating information from observed to unobserved variables via sequential probabilistic inference. However, the set of assumptions is drastically reduced. It does not make any assumptions about distributions of $x_i^0$ and does not strictly require the Markov assumption about conditional independence between variables. Instead, the network can (but not has to) be conditioned on all other variables, utilizing compression capabilities of neural networks. Due to the nature of generative models, however, the formulation does not allow for explicit representation of probabilities but allows only sampling.

### 3.1.1. SAMPLING IN CONTINUOUS DOMAINS

We first introduce our formulation for sampling a single continuous random variable. Denoising generative models are the established state-of-the-art for learning continuous data distributions (Dhariwal & Nichol, 2021). While there are multiple different derivations like score matching (Song & Ermon, 2019), DDPM (Ho et al., 2020), or the recently more popular (conditional) flow matching (Lipman et al., 2023), they all share the same idea of learning a step-wise mapping of scalar or higher-dimensional continuous variables $x^t$ from a simple and known Gaussian distribution $x^1 \sim \mathcal{N}(0, I)$ to the complex and unknown data distribution $x^0 \sim q$. With small enough step sizes $t' - t$ (Ho et al., 2020), each step itself can be modeled by sampling from a Gaussian distribution (with possibly zero variance)

$$x^t \sim \mathcal{N}(\mu_\theta(x^{t'}), \Sigma_\theta(x^{t'})), \quad (3)$$

parameterized by a neural network with weights $\theta$, trained to denoise noisy versions of samples from the data distribution. Training examples are constructed by interpolating between samples $x^0 \sim q$ and Gaussian noise $\epsilon \sim \mathcal{N}(0, I)$ as

$$x^t = a(t)x^0 + b(t)\epsilon, \quad (4)$$

where $t \in [0, 1]$ is a continuous level of noise and $a, b : [0, 1] \mapsto [0, 1]$ define interpolation weights. While DDPM (Ho et al., 2020) limits itself to a subset of possible noise schedules, alternatives like rectified flows were recently explored as general Gaussian probability paths in the context of conditional flow matching (Lipman et al., 2023).

In this work, we construct a specific reverse distribution, i.e., mean and variance in Eq. 3 for arbitrary noise schedules $a, b$ with its marginal distribution satisfying Eq. 4, similar as but more general than DDIM (Song et al., 2021). We provide all details and proofs in Appendix A. As a result, our formulation can combine non-diffusion noise schedules such as from rectified flows (Liu et al., 2023) ($a(t) = 1 - t, b(t) = t$) with stochastic sampling as in DDPM and follow-ups such as learning the variances of the reverse distributions (Nichol & Dhariwal, 2021).

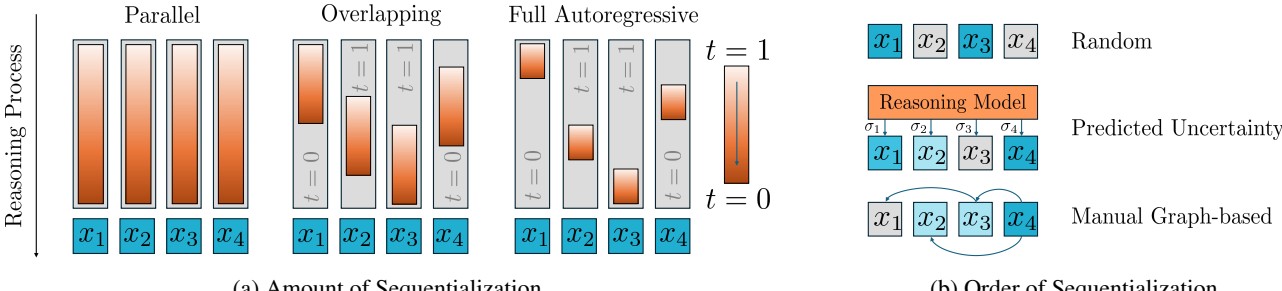

(a) Amount of Sequentialization

(b) Order of Sequentialization

Figure 3: **Investigated Degrees of Freedom.** **(a)** The SRM framework allows to define different amounts of sequentialization, i.e. parallel generation, autoregressive generation or mixtures with varying overlap, modeled by differing levels of noise on individual variables at the same time. **(b)** Further, it provides different options to define the order of sequentialization, allowing random ordering, a greedy heuristic based on predicted uncertainty and manually-defined graphs.

### 3.1.2. SAMPLING IN SPATIAL DOMAINS

We model our spatial reasoning problems as inferring sets of continuous random variables, unifying previous denoising generative approaches and spatially autoregressive methods.

On the one hand, existing diffusion- and flow-based models like popular image generators (Rombach et al., 2022) reason only along the noise dimension by assuming single, shared noise levels for all variables at any point in time during the denoising process. By setting $t_1 = ... = t_n$ and $t'_1 = ... = t'_n$ in Eq. 2, this special case is subsumed by our framework. We hypothesize potential limitations of this approach in the presence of complex dependencies between spatially distinct variables, as in the case of MNIST Sudoku (Fig.2).

On the other hand, spatially autoregressive approaches like MAR (Li et al., 2024) learn to sample a next variable $x_i$ conditioned on the history of previously sampled variables $x_j$ for $j \in \mathcal{J}$ and $\{\{i\}, \mathcal{J}, \mathcal{K}\}$ being a partitioning of $\{1, ..., n\}$. Our framework covers this specific way of sampling by setting $t_j = t'_j = 0$ for $j \in \mathcal{J}$ and $t_k = t'_k = 1$ for $k \in \mathcal{K}$. Although the application of the chain-rule can decompose the complex joint distribution of variables into potentially simpler conditional distributions, we argue that potential improvements depend on the order, which is usually simply set to be random or a raster scan.

These two special instances depict the extreme cases of our framework. Given that both have their own strengths and weaknesses, we aim to explore the space in between. Therefore, we train spatial reasoning models to jointly denoise multiple variables but with individual noise levels.

### 3.2. Training

We parameterize a SRM as a noise prediction network $\epsilon_\theta$ that is trained to regress the ground truth noise $\epsilon \sim \mathcal{N}(\mathbf{0}, \mathbf{I})$

$$\mathcal{L}_\mu = \mathbb{E}_{\epsilon, \mathbf{t}, \mathbf{x}} \|\epsilon_\theta(\mathbf{x^t}) - \epsilon\|^2, \tag{5}$$

where $\mathbf{t} := (t_1, ..., t_n)$, $\mathbf{x} := (x_1, ..., x_n)$, and $\mathbf{x^t} := (x_1^{t_1}, ..., x_n^{t_n})$. Additionally, we let the network predict the variance of the reverse process in Eq. 3, optimized for the variational lower bound (Nichol & Dhariwal, 2021). As shown in previous works (Esser et al., 2024), the training of a denoising network is very sensitive to the sampling of noise levels. This is especially true for our setting of individual noise levels for a possibly large number of variables.

### 3.2.1. NOISE LEVEL SAMPLING

Sampling noise levels during training of diffusion models allows the model to learn denoising images of all noise levels $t$. In SRMs, the image patches also act as conditioning to other patches, so we need to ensure that the whole $t$ distribution is similar to what happens during inference.

At the beginning of inference, all patches $p_i$ start with $t_i = 1$, so the mean noise level is initially $\bar{t} = 1$. Similarly, at the end of inference, all patches reach $t_i = 0$, giving $\bar{t} = 0$. For diffusion models with linear time schedules, or our SRM with a *parallel* schedule, the mean noise level decreases by a fixed amount per inference step $\Delta \bar{t} = -\frac{1}{N}$, where $N$ is the number of inference steps. In the autoregressive generation, a similar relation holds $\Delta \bar{t} = -\frac{1}{M \cdot P}$, where $M$ is the number of steps per patch, and $P$ is the number of patches. In both cases, $\Delta \bar{t}$ does not depend on the noise level, which suggests that with a sufficient number of inference steps, $\bar{t}$ should ideally follow a uniform distribution. See Fig. 9 in appendix for visualization.

*Diffusion Forcing* (Chen et al., 2024a) suggests to independently sample a noise level vector $\mathbf{t} \sim \mathcal{U}([0, 1]^n)$, ensuring a uniform marginal distribution, $t_i \sim \mathcal{U}(0, 1)$. However, this leads to the mean $\bar{t}$ following a Bates distribution, which is highly concentrated around $0.5$ – significantly different from the distribution encountered during inference (c.f. Fig. 4a). As a result, a trained model is undertrained for early (and late) inference steps, where most patches have noise levels close to 1 (and 0), reducing performance (c.f. Sec. 4.6).

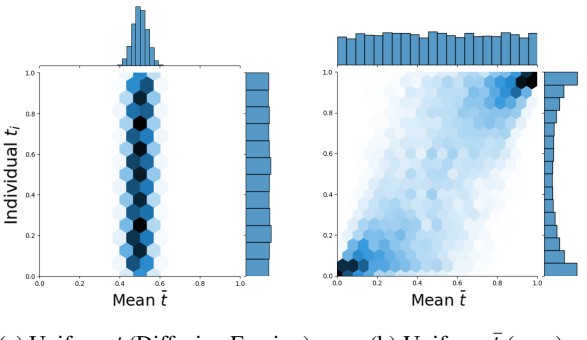

(a) Uniform $t$ (Diffusion Forcing)    (b) Uniform $\bar{t}$ (ours)

Figure 4: **Noise Level Sampling.** We visualize the joint distribution of noise levels $t_i$ and mean noise levels $\bar{t}$ for (a) *Diffusion Forcing* training and (b) our Uniform $\bar{t}$ approach. In *Diffusion Forcing*, the distribution of $t_i$ is correctly modeled but $\bar{t}$ is overly concentrated around $\bar{t} = 0.5$, which does not reflect the distribution seen during inference. Our Uniform $\bar{t}$ addresses this issue, though it introduces some bias in the distribution of individual $t_i$ which we address by per-patch loss weighting.

To address this issue, we introduce a two-step sampling strategy called Uniform $\bar{t}$. We first sample $\bar{t} \sim \mathcal{U}(0,1)$, and then generate $\mathbf{t} \sim p(\mathbf{t} \mid \bar{t})$ using our *recursive allocation sampling* algorithm (Appendix C.1). The algorithm allows control of the *sharpness* of the $p(\mathbf{t} \mid \bar{t})$ – the spread of individual $t_i$ around $\bar{t}$. This makes it adaptable to different inference scenarios flexible for different inference settings. For example, in the parallel generation, $t_i$ should stay close to $\bar{t}$ (high *sharpness*). In contrast, autoregression benefits from greater cross-patch noise level variance.

Uniform $\bar{t}$ sampling preserves the uniform mean distribution but oversamples individual patch noise levels close to $t_i = 0$ and $t_i = 1$ (c.f. Fig. 4b). This occurs because if $\bar{t} \approx 0$ is sampled, all $t_i$ must be close to 0 and similarly for $\bar{t} \approx 1$, $\forall_i t_i \approx 1$. On the other hand, for $\bar{t} \approx 0.5$, the $t_i$ samples can take any value within the range. To counteract this bias towards sampling $t_i \approx 0$ or $t_i \approx 1$, we introduce per-patch loss weights $w_t = \frac{1}{p(t)}$, where $p(t)$ is empirically estimated.

### 3.2.2. UNCERTAINTY ESTIMATION

To sample variables in a meaningful order, we propose to train the denoising network to predict the uncertainty in its noise prediction. As common in heteroscedastic uncertainty estimation (Seitzer et al., 2022), we model the uncertainty as standard deviation $\sigma_\theta(\mathbf{x^t})$ and minimize the negative log-likelihood (NLL) of the ground truth noise $\epsilon$

$$\mathcal{L}_\sigma = \mathbb{E}_{\epsilon, \mathbf{t}, \mathbf{x}} - \log \mathcal{N}(\epsilon | \epsilon_\theta(\mathbf{x^t}), \sigma_\theta(\mathbf{x^t})^2 I). \quad (6)$$

To avoid deteriorating the noise prediction, we only train the uncertainty estimation with this loss and a small weight.

### 3.3. Sampling

A trained SRM can be used for various sampling techniques and therefore benchmark them fairly using a single model. We investigate different amounts and orders of sequentialization for the denoising of spatial variables (see Fig. 3).

Regarding the amount of sequentialization, we explore parallel generation as usual in denoising generative models, the other extreme of fully autoregressive sampling, and a mixture achieved by overlapping the denoising step intervals of the different variables by a variable amount.

Besides the amount of sequentialization, we further compare different orders of variables. Leveraging the estimated uncertainty in the noise prediction from Sec. 3.2.2, we propose to adaptively sample the next variable with the lowest uncertainty. This strategy follows the assumption that more uncertain variables can benefit from stronger conditioning of many clean variables later during sampling. For some tasks, we know the dependency structure between spatial variables in advance. Considering our Sudoku example, a cell depends directly on the other cells in its row, column, and 3x3 block. Encoding this information in form of an adjacency matrix, we propose to propagate certainty based on each variable's noise level to sample the most certain variable next. We include a random order as further baseline.

## 4. Experiments

We evaluate SRMs for reasoning on three new benchmark datasets that we introduce in Sec. 4.1. We consider images split into patches as our sets of continuous random variables and use 2D UNets for denoising as described in Sec. 4.2. Besides our main findings detailed in Sec. 4.3 to 4.5, we provide additional ablations in Sec. 4.6.

### 4.1. Benchmark Datasets

We introduce three different datasets to quantify reasoning capabilities. They are aimed at different aspects to be tested. The *MNIST Sudoku* dataset captures complex (NP-hard) dependencies that need to be understood. The *Even Pixels* dataset is an easier task that can be solved in a greedy fashion. Finally, we introduce the *Counting Polygons / Stars FFHQ* dataset, which moves closer to real-world images.

**MNIST Sudoku Dataset**    We create a dataset based on one million correct Sudoku instances by randomly sampling MNIST representatives for each cell (see Fig. 2a for examples). The classification of Sudoku images into correct and incorrect ones is then done via an application of a pre-trained MNIST digit classifier MLP on each cell and a check whether all Sudoku rules are fulfilled. To avoid classification errors distorting our reasoning results, we limit the set of MNIST digit representatives to the top 1000 training examples per class of the classifier w.r.t. its confidence.

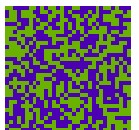 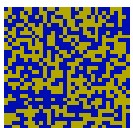 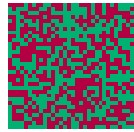 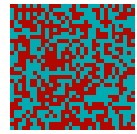 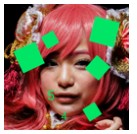 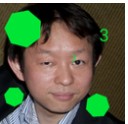 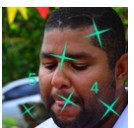 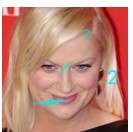

(a) Even Pixels dataset            (b) Counting Polygons / Stars FFHQ dataset

Figure 5: **Two Further Datasets.** In addition to MNIST Sudoku, we introduce two further datasets. (a) The *Even Pixels* dataset requires the model to generate images with an equal number of pixels per color. (b) *Counting Polygons / Stars FFHQ* requires that all images contain two digits: the number of objects and the number of polygon edges or star points, respectively. We introduce real-world complexity by placing the synthetic objects over FFHQ samples as backgrounds.

For testing, we use a held-out dataset split of valid Sudokus and apply random masking of cells with the number of masked ones randomly sampled from the intervals $[1, 27]$, $[28, 54]$, and $[55, 81]$, resulting in three levels of difficulty easy, medium, and hard, respectively. As metrics, we consider accuracy as well as the sum of $L_1$-distances of row-, column-, and block-wise digit histograms to the all ones vector (zero if correct), averaged over all test examples.

The game Sudoku involves complex spatial dependencies, whereas the distribution of individual cells being MNIST numbers is simple to fit by a generative model. Note that the general task of solving Sudokus for a variable grid size is NP-complete and that depending on the level of masking, there can be multiple valid solutions resulting in uncertainty. We further highlight the difficulty of even a discrete Sudoku version for state-of-the-art LLMs in Appendix H.

For our graph-based order for sequentialization during sampling, we encode the direct dependencies between each cell and its neighbors within a row, column, and block as the adjacency matrix. Additionally, we provide results for oracle algorithms that directly solve discrete Sudokus by randomly sampling a number for the next cell from all possible digits avoiding collisions with neighbors if possible. The order of cells is chosen either randomly or in a greedy fashion, with the next cell being the one with the largest number of already sampled neighbors in the constructed graph.

**Even Pixels Dataset** We construct a dataset of images where each pixel is assigned one of two opposite colors (c.f. Fig. 5a). The number of pixels of each color is always equal. The task requires a model to learn this implicit constraint without explicit supervision. We detail our evaluation procedure in Appendix D.

**Counting Polygons / Stars FFHQ Dataset** We also introduce two datasets in which each image contains a set of randomly positioned polygons or stars and numbers (see Fig. 5b), where: The number of synthetic objects, $N$, is randomly sampled from $\{1, \ldots, 9\}$, each polygon / star has $K \in \{3, \ldots, 7\}$ vertices or $K \in \{2, \ldots, 9\}$ points, respectively, and the numbers $N$ and $K$ are explicitly present in the image. The model's task is to generate images following this rule and hence to capture the relationship between the

number of polygons / stars, vertices / points, and the digits.

To introduce real-world complexity, we place these synthetic objects over backgrounds sampled from the FFHQ dataset. This forces the model to notice and learn meaningful spatial dependencies in a setting much closer to the real world. We train ResNet classifiers to estimate correctness of generated samples and verify their capability on held-out validation splits. More details are given in Appendix E.

### 4.2. Experimental Setup

We compare SRMs in pixel space with standard denoising diffusion models that are equivalent in architecture and training up to the sampling of individual spatial variables, which are chosen to be image patches of task-specific sizes for our visual reasoning tasks. Since our MNIST Sudoku benchmark can be seen as an instance of inpainting, the baseline diffusion model is additionally conditioned on the incomplete Sudoku and its mask via concatenation in the input. Note that this is not necessary for SRMs, as already given variables are identified by the noise level zero. If not stated otherwise, we use stochastic sampling similar to DDPM with a total of $1000$ steps (network evaluations) independent of the method of sequentialization during sampling. Therefore, the computational cost is equal for all sampling methods resulting in a fair comparison. This also means that there are fewer denoising steps for individual variables the higher the degree of sequentialization (lower overlap). SRMs are agnostic to both noise schedules and denoiser architectures. For the main paper, we use rectified flows (Liu et al., 2023) and lightweight versions of widely established 2D UNets with spatial attention in low-resolution layers to avoid results being dominated by too extreme overparameterization. We report additional results with the cosine noise schedule (Nichol & Dhariwal, 2021) and Diffusion Transformers (DiT) (Peebles & Xie, 2023) in Appendix G, describe the exact sampling process in Appendix A, and provide implementation details in Appendix F.

### 4.3. MNIST Sudoku Results

We provide a quantitative comparison of SRM with different sampling strategies, a standard diffusion model as baseline, and the task-specific discrete oracle algorithms in Tab. 1.

| Model | Sampling | Easy | | Medium | | Hard | |
|---|---|---|---|---|---|---|---|
| | | L1↓ | Acc↑ | L1↓ | Acc↑ | L1↓ | Acc↑ |
| Diffusion Model | Parallel | 0.032 | 0.994 | 3.924 | 0.536 | 14.120 | 0.008 |
| SRM (Ours) | Parallel | 0.012 | 0.998 | 3.312 | 0.590 | 19.156 | 0.010 |
| | Predicted Order w/o Overlap | 0.012 | 0.998 | **1.616** | **0.754** | **3.212** | **0.516** |
| | Predicted Order + Overlap | **0.000** | **1.000** | 1.816 | 0.716 | 4.340 | 0.422 |
| | Random Order w/o Overlap | 0.012 | 0.998 | 3.828 | 0.568 | 12.612 | 0.024 |
| | Random Order + Overlap | **0.000** | **1.000** | 4.172 | 0.542 | 13.896 | 0.020 |
| | Graph-based Order w/o Overlap | 0.020 | 0.996 | 2.544 | 0.662 | 5.816 | 0.266 |
| | Graph-based Order + Overlap | **0.000** | **1.000** | 3.240 | 0.576 | 6.036 | 0.238 |
| Oracle | Random | - | 0.751 | - | 0.059 | - | 0.000 |
| | Greedy | - | 1.000 | - | 0.987 | - | 0.672 |

Table 1: **MNIST Sudoku Quantitative Results.** While standard DDPM diffusion models fail to solve the hard Sudoku cases (accuracy near $0\%$, SRMs achieve an accuracy of $> 50\%$). Interestingly, the Sudoku task particularly profits from predicted uncertainty, which heavily outperforms random sequentialization order and also the graph-based ordering. We provide the accuracy of a combinatorial Sudoku solver using a greedy strategy (without backtracking) as reference (Oracle).

While the (inpainting) diffusion model that does fully parallel denoising of all patches is able to solve *easy* examples with a large number of given cells, its performance deteriorates drastically with weaker conditioning. SRM with parallel sampling behaves similar, indicating that this sampling strategy is inappropriate for the Sudoku task with complex dependencies, independent of the network training. For all spatially autoregressive sampling methods, having an overlap between denoising intervals of patches sampled one after the other, decreases the performance compared to the full autoregressive extreme. This highlights the importance of spatial sequentialization for reasoning.

Furthermore, our results clearly show that the order of sequentialization matters. SRM with a random order achieves mediocre performance. As the model always sees the entire (noisy) Sudoku, the network learns that the distribution of the currently denoised patch does not only depend on all previously denoised cells but also on the possible solutions for all future ones, explaining the advantage over the random oracle (which fails completely). Using task-specific knowledge about spatial dependencies in form of a general graph helps to significantly improve the order of sequentialization and as a result sampling from the correct distribution.

However, we achieve the best performance by a large margin with the task-agnostic order based on the predicted uncertainty as described in Sec. 3.2.2. An example for such a sampling process is visualized in Fig. 7. We depict three pairs of close steps at the start, middle, and end of sampling. By following an exclusion procedure for the digit one in the middle right block using clean conditionings only, we understand why the model correctly predicted the lowest uncertainty for that patch at the start. Moving to the middle case, the chosen cell is also completely determined by the conditioning, which only becomes visible if one takes into

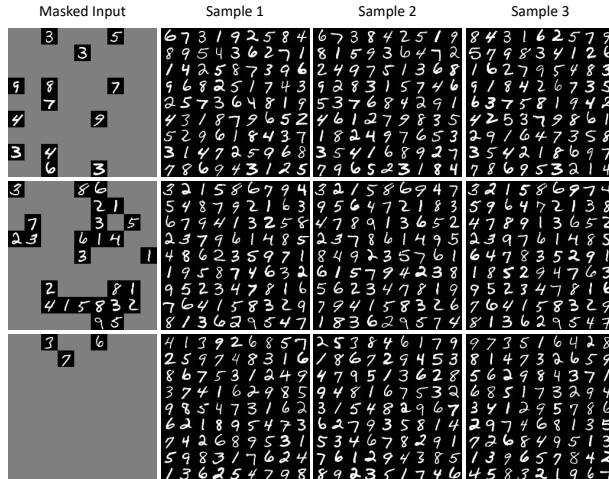

Figure 6: **Sample Diversity.** SRMs can sample multiple different, correct solutions given an incomplete observation.

account other cells that are not yet denoised but already determined by applying the rules of Sudoku. We encourage the reader to verify this for themselves. In the end, we have a case of multiple valid solutions, from which SRMs, being generative models, are able to sample. We further demonstrate the sample diversity for multiple incomplete Sudokus in Fig. 6. We provide videos on our project website that visualize the full sampling process for multiple examples.

Overall, we improve the correctness of samples in the hard setting from $0.8\%$ to $51.6\%$ (cf. Tab. 1) using the same trained model, only varying the sampling strategy.

### 4.4. Even Pixels Results

Tab. 2 shows the quantitative comparison of our best settings for all sampling methods. We can see a clear gap between

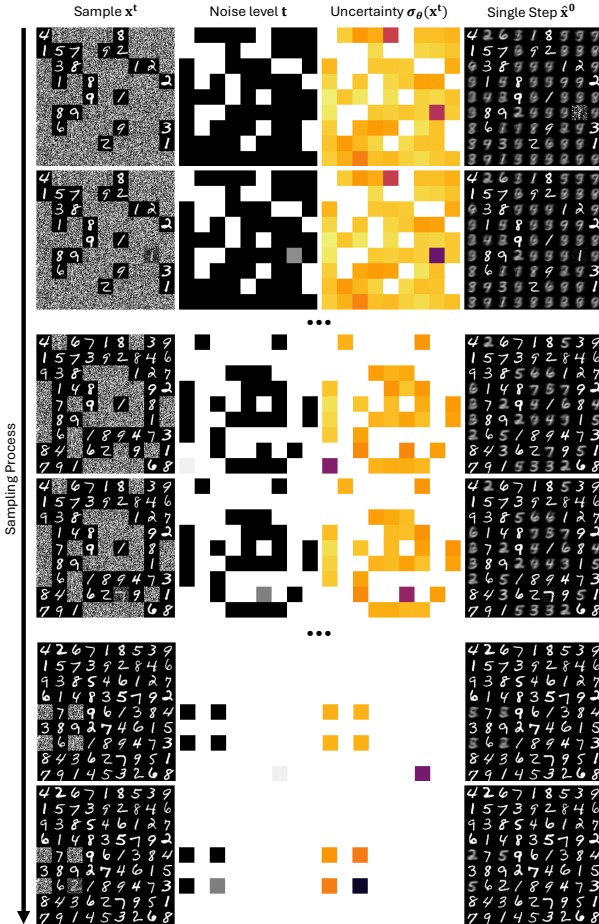

Figure 7: **Sequentialization With Predicted Order.** We visualize the sampling process with the current sample $\mathbf{x^t}$, noise level $\mathbf{t}$, estimated uncertainty (darker = lower) $\sigma_\theta(\mathbf{x^t})$, and the single step to $\mathbf{t} = 0$ result $\hat{\mathbf{x}}^0$. SRMs are able to reason over spatial variables by capturing complex dependencies (first two blocks) and handle uncertainty, e.g., in the case of multiple valid solutions as in the last block.

| Samling | #E↓ | Acc↑ |
|---|---|---|
| Diffusion Model | 1.270 | 0.250 |
| Ours, Parallel | 5.184 | 0.054 |
| Ours, Predicted Order + Overlap | **0.534** | **0.518** |
| Ours, Random Order + Overlap | 0.584 | 0.476 |

Table 2: **Even Pixels Experiments.** SRMs clearly outperform a standard diffusion model in this task. Random order and predicted order perform similar, with a slight advantage for the predicted one. See Fig. 8 for overlap analysis.

the diffusion baseline and SRM. While the diffusion model must learn to balance pixel colors evenly on a global level, sequentially eliminating differences in pixel counts can sim-

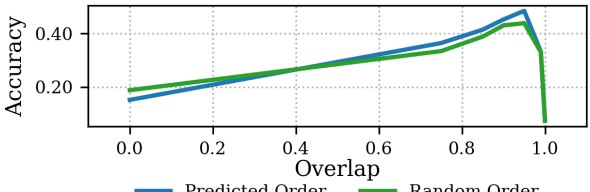

Figure 8: **Even Pixel Accuracy for Different Overlap.** Larger overlap is better for generating an even number of pixels. However, when approaching the parallel setting at 1.0, the performance heavily drops.

plify the task. This behavior can be further encouraged by lowering the *sharpness* of our noise level sampling during training (cf. Sec. 3.2.1), i.e., slightly biasing the SRM training towards spatially autoregressive generation.

Unlike for the MNIST Sudoku experiment, we find overlapping denoising of variables to be beneficial and further visualize this relationship in Fig. 8. For both predicted and random order, the accuracy gradually increases with a higher overlap until a sweet spot at 0.95, after which it falls quickly. This shows that depending on the data distribution, mixtures of parallel and sequential generation enabled by SRMs can be advantageous. We can see a small, consistent advantage of the predicted uncertainty order over the random one.

### 4.5. Counting Polygons / Stars Results

For the final Counting Polygons / Stars on FFHQ datasets, we provide the quantitative results for sample accuracy (fulfillment of dataset rule) and consistency of objects (same number of polygon vertices / star points) in Tab. 3. SRM outperforms the diffusion model for almost all sampling methods. This indicates potential of our training with individual noise levels, as this form of spatial disentanglement might be generally beneficial for certain data distributions.

For the Counting Polygons dataset variant, there is no clearly winning sampling strategy, unlike for the other two benchmarks. We attribute this to two main differences of the dataset. First, using real images for backgrounds makes the task significantly more complex, as the denoising objective becomes less sensitive to the dependencies between numbers and polygons and capacity of the denoising network is spent for fitting the distribution of FFHQ faces. Secondly, the simultaneous generation of matching numbers and polygons can be approached in a coarse-to-fine manner with the numbers being more high-frequency details compared to larger low-frequency polygons. For MNIST Sudoku, all numbers have the same sizes such that, during sampling with a parallel denoising strategy, the commitment to individual digits has to happen at similar points in time. Due to the spatial dependencies of Sudoku, this is suboptimal and a

| Dataset | Polygons | | Stars | |
| --- | --- | --- | --- | --- |
| Sampling | Acc↑ | Con↑ | Acc↑ | Con↑ |
| Diffusion Model | 0.132 | 0.990 | 0.070 | 0.544 |
| Ours, Parallel | 0.166 | 0.956 | 0.034 | 0.576 |
| Ours, Predicted Order w/o Overlap | 0.144 | 0.994 | 0.076 | 0.844 |
| Ours, Predicted Order + Overlap | 0.142 | 0.986 | **0.150** | 0.888 |
| Ours, Random Order w/o Overlap | 0.164 | **0.998** | 0.080 | 0.872 |
| Ours, Random Order + Overlap | **0.186** | 0.964 | 0.104 | **0.938** |

Table 3: **Counting Polygons / Stars FFHQ Experiments.** Although this task is generally hard and not solved well, sequentialization is beneficial, especially for consistency.

| Sampling / $t$-Sampling | Uniform $t$ | Ours |
| --- | --- | --- |
| Parallel | 0.000 | **0.008** |
| Predicted Order w/o Overlap | 0.018 | **0.516** |
| Predicted Order + Overlap | 0.008 | **0.422** |
| Random Order w/o Overlap | 0.000 | **0.024** |
| Random Order + Overlap | 0.000 | **0.020** |
| Graph-based Order w/o Overlap | 0.012 | **0.266** |
| Graph-based Order + Overlap | 0.010 | **0.238** |

Table 4: **Noise Level Sampling Ablation.** Ablation performed on MNIST Sudoku hard. Our strategy for sampling $t$'s during training is crucial for all strategies to work.

spatially autoregressive strategy together with a good order can commit to the digits in a cell-by-cell fashion. For the Counting Polygons dataset, the numbers are high-frequency details compared to larger polygons. As a result, a coarse-to-fine generation with the diffusion baseline can first commit to a number of polygons of a certain vertex count and then generate matching digits.

We further validate this hypothesis by replacing the low-frequency polygons with stars that are composed of higher-frequencies, moving the "point of commitment" to numbers and stars closer together in time. For this dataset variant, we observe that the accuracy of the diffusion baseline decreases significantly compared to the version with polygons (7% vs 13.2%), while sequential sampling with overlap and predicted order maintains the same performance. More interestingly, for parallel generation, we noticed hallucinations of samples with stars having inconsistent numbers of points (cf. consistency column). For sequential sampling, the model can replicate stars after the first one has been generated, whereas in parallel sampling, the decisions over the number of points for all stars are again closer in time. We hypothesize that this behavior is not visible for polygons because of differences in terms of frequencies, with the diffusion model having more "time to correct itself" for low-frequency polygons, as their generation starts earlier than for the high-frequency stars.

### 4.6. Ablations

In Tab. 4, we provide the results of an ablation of the noise level sampling from Sec. 3.2.1 on the hard difficulty of the MNIST Sudoku dataset. Enforcing a uniformly distributed mean during training is essential for all sampling methods. While independent uniform sampling of noise levels might be good enough for a very small number of variables as in Diffusion Forcing (Chen et al., 2024a), high and low mean noise levels quickly become severely undersampled when we increase it. As these two cases represent start and end points at test time, the input to the network is immediately out of the training distribution during sampling. We provide additional ablations w.r.t. stochasticity of inference,

denoiser architecture, noise schedule, and the sharpness hyperparameter (cf. Sec. 3.2.1) in Appendix G.

## 5. Conclusion

We introduced Spatial Reasoning Models (SRMs), a framework that allows to reason over sets of continuous, spatial variables. Our framework allows parameterization for several different strategies of sequentialization and generation order. Of particular interest is an automatic order prediction based on uncertainty. The SRM framework also is agnostic to the employed denoising formulation and parameterization, and allows to use a wide variety of different denoising architectures. We introduced three benchmark tasks, which demonstrate that SRMs significantly outperform standard diffusion models in higher level reasoning tasks.

**Future Directions**  While we made significant steps forward, our models are still far away from solving the given tasks in an optimal fashion. However, we believe that the presented paradigm is able to achieve even better results. A particular topic for future research can be more advanced strategies for automatic prediction of generation order. It is an interesting venture to further investigate what makes a distribution difficult to generate for a diffusion model. A potential difficulty metric could help optimizing for an optimal order. Also, we speculate that *backtracking*-like strategies that allow to increase noise levels again hold much potential.

## Impact Statement

This paper presents fundamental research with the goal of advancing reasoning capabilities of generative models. There are many potential societal consequences of advances in this direction further down the road, however, none are immediate enough to be specifically highlighted here.

## Acknowledgements

This work was partially supported by the Saarland/Intel Joint Program on the Future of Graphics and Media. We thank Thomas Wimmer for proofreading and helpful discussions.

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

The appendix is structured as follows. First, the detailed unified denoising framework is given in Sec. A. Then, graph-based sampling strategies are given in Sec. B. It follows a detailed description of the Uniform $\bar{t}$ sampling in Sec. C and dataset descriptions in Sec. D and Sec. E. The appendix concludes with further details about experimental setup in Sec. F. Please also take a look at our project website with, among other things, videos showing Sudoku reasoning over time.

# A. Generation Process

**Notation**   Given a noise schedule $a, b : [0, 1] \mapsto [0, 1]$ for continuous levels of noise $t \in [0, 1]$, we define $a_t := a(t), b_t := b(t)$ for short notation. Furthermore, as we consider a single (possibly higher-dimensional) continuous random variable for simplicity, we write $x_t$ for the variable at noise level $t$ instead of using superscript as in the main paper.

**Reverse Distribution**   Similarly to DDIM (Song et al., 2021), we define a family of inference distributions indexed by the function $\sigma : \{(t^*, t) \in [0, 1]^2 \mid t^* < t\} \mapsto \mathbb{R}_{\geq 0}$ with $\sigma_{t^*,t} := \sigma(t^*, t) \leq b_{t^*}$ for a fixed-sized schedule of continuous noise levels $t_i \in [0, 1]$ with $1 \leq i \leq N$, $N \in \mathbb{N}_{\geq 2}$, $t_1 = 0$, $t_N = 1$, and $t_{i-1} < t_i$:

$$q_\sigma(x_{0:1}|x_0) := q_\sigma(x_1|x_0) \prod_{i=2}^{N} q_\sigma(x_{t_{i-1}}|x_{t_i}, x_0), \tag{7}$$

where $q_\sigma(x_1|x_0) = \mathcal{N}(a_1 x_0, b_1^2 I)$ and for all $0 \leq t_{i-1} < t_i \leq 1$:

$$q_\sigma(x_{t_{i-1}}|x_{t_i}, x_0) = \mathcal{N}\left(a_{t_{i-1}} x_0 + \frac{x_{t_i} - a_{t_i} x_0}{b_{t_i}} \sqrt{b_{t_{i-1}}^2 - \sigma_{t_{i-1},t_i}^2}, \sigma_{t_{i-1},t_i}^2 I\right). \tag{8}$$

We show that the choice of the mean ensures the desired marginal distribution $q_\sigma(x_t|x_0) = \mathcal{N}(a_t x_0, b_t^2 I)$.

*Proof.* By construction, the statement holds for $t = 1$. Let $0 \leq t^* < 1$, then we have

$$q_\sigma(x_t|x_0) = \mathcal{N}(a_t x_0, b_t^2 I) \tag{9}$$

$$q_\sigma(x_{t^*}|x_t, x_0) = \mathcal{N}\left(a_{t^*} x_0 + \frac{x_t - a_t x_0}{b_t} \sqrt{b_{t^*}^2 - \sigma_{t^*,t}^2}, \sigma_{t^*,t}^2 I\right). \tag{10}$$

From (Bishop & Nasrabadi, 2006) (2.115), we have that $q_\sigma(x_{t^*}|x_0)$ is a normal distribution $\mathcal{N}(\mu, \Sigma)$ with:

$$\mu = a_{t^*} x_0 + \frac{a_t x_0 - a_t x_0}{b_t} \sqrt{b_{t^*}^2 - \sigma_{t^*,t}^2} = a_{t^*} x_0 \tag{11}$$

$$\Sigma = \sigma_{t^*,t}^2 I + \frac{b_{t^*}^2 - \sigma_{t^*,t}^2}{b_t^2} b_t^2 I = b_{t^*}^2 I \tag{12}$$

Therefore, $q_\sigma(x_t|x_0) = \mathcal{N}(a_t x_0, b_t^2 I)$ holds for all $t \in [0, 1]$. $\qquad \square$

Via simple variable substitution $\epsilon = \frac{z_t - a_t x_0}{b_t}$ and $x_0 = \frac{z_t - b_t \epsilon}{a_t}$, we obtain the corresponding distributions for alternative conditionings:

$$q_\sigma(x_{t^*}|\epsilon, x_0) = \mathcal{N}\left(a_{t^*} x_0 + \epsilon \sqrt{b_{t^*}^2 - \sigma_{t^*,t}^2}, \sigma_{t^*,t}^2 I\right) \tag{13}$$

$$q_\sigma(x_{t^*}|\epsilon, x_t) = \mathcal{N}\left(\frac{a_{t^*} x_t}{a_t} + (\sqrt{b_{t^*}^2 - \sigma_{t^*,t}^2} - \frac{a_{t^*} b_t}{a_t})\epsilon, \sigma_{t^*,t}^2 I\right), \tag{14}$$

where we use Eq. 13 as the posterior distribution approximated by minimizing the variational lower bound and Eq. 14 for the following definition of our generative process by replacing the ground truth epsilon with the prediction of the denoising neural network

$$q_{\theta,\sigma}(x_{t^*}|, x_t) = \mathcal{N}\left(\frac{a_{t^*} x_t}{a_t} + (\sqrt{b_{t^*}^2 - \sigma_{t^*,t}^2} - \frac{a_{t^*} b_t}{a_t})\epsilon_\theta(x_t), \Sigma_{t^*,t}\right), \tag{15}$$

where $\Sigma_{t^*,t}$ can be either fixed as in DDPM (Ho et al., 2020), e.g., to equivalent upper $\Sigma_{t^*,t} = \frac{b_t}{b_{t^*}} \sigma_{t^*,t}^2 I$ and lower bounds $\Sigma_{t^*,t} = \sigma_{t^*,t}^2 I$ depending on $q(x_0)$ being isotropic noise or a delta function, or learned by the network as an interpolation between these two optimized for the VLB, as proposed by (Nichol & Dhariwal, 2021) for DDPM. In all experiments of the paper, we choose the last option.

**Choice of $\sigma$** For the choice of the posterior standard deviation $\sigma_{t^*,t}$, we follow DDIM (Song et al., 2021) and consider the following special case. If and only if $\sigma(t^*, t) = b_{t^*}\sqrt{1 - (a_t b_{t^*}/(a_{t^*} b_t))^2}$ for $0 \leq t^* < t \leq 1$, the forward process becomes Markovian, i.e., $q_\sigma(x_t|x_{t^*}, \mathbf{x}_0) = q_\sigma(x_t|x_{t^*})$.

*Proof.* Using the abbreviation $\sigma$ for $\sigma(t^*, t)$, we have

$$q_\sigma(x_t|x_0) = \mathcal{N}(a_t x_0, b_t) \tag{16}$$

$$q_\sigma(x_{t^*}|x_t, x_0) = \mathcal{N}\left(a_{t^*} x_0 + \frac{x_t - a_t x_0}{b_t}\sqrt{b_{t^*}^2 - \sigma^2}, \sigma^2 I\right) \tag{17}$$

From (Bishop & Nasrabadi, 2006) (2.116), we have $q_\sigma(x_t|x_{t^*}, x_0) = \mathcal{N}(\mu, \Sigma)$ with:

$$\Sigma = \left(\frac{1}{b_t^2} + \frac{b_{t^*}^2 - \sigma(t^*,t)^2}{b_t^2}\frac{1}{\sigma(t^*,t)^2}\right)^{-1} I = \frac{b_t^2 \sigma(t^*,t)^2}{b_{t^*}^2} I \tag{18}$$

$$\mu = \frac{b_t^2 \sigma^2}{b_{t^*}^2}\left[\frac{\sqrt{b_{t^*}^2 - \sigma^2}}{b_t \sigma^2}\left(x_t + \left(\frac{a_t}{b_t}\sqrt{b_{t^*}^2 - \sigma^2} - a_{t^*}\right)x_0\right) + \frac{a_t x_0}{b_t^2}\right] \tag{19}$$

We can see that $\mu$ becomes independent of $\mathbf{x}_0$ for the case:

$$\frac{\sqrt{b_{t^*}^2 - \sigma^2}}{b_t \sigma^2}\left(\frac{a_t}{b_t}\sqrt{b_{t^*}^2 - \sigma^2} - a_{t^*}\right) + \frac{a_t}{b_t^2} = 0 \tag{20}$$

$$\Longleftrightarrow \frac{a_t}{b_t \sigma^2}(b_{t^*}^2 - \sigma^2) - \frac{a_{t^*}}{\sigma^2}\sqrt{b_{t^*}^2 - \sigma^2} + \frac{a_t}{b_t} = 0 \tag{21}$$

$$\Longleftrightarrow a_t(b_{t^*}^2 - \sigma^2) - a_{t^*} b_t \sqrt{b_{t^*}^2 - \sigma^2} + a_t \sigma^2 = 0 \tag{22}$$

$$\Longleftrightarrow a_t b_{t^*}^2 = a_{t^*} b_t \sqrt{b_{t^*}^2 - \sigma^2} \tag{23}$$

$$\Longleftrightarrow \sigma = b_{t^*}\sqrt{1 - \left(\frac{a_t b_{t^*}}{a_{t^*} b_t}\right)^2} \tag{24}$$

$\square$

We define $\sigma_\eta(t^*, t) := \eta \cdot b_{t^*}\sqrt{1 - (a_t b_{t^*}/(a_{t^*} b_t))^2}$ for $\eta \in [0, 1]$ as the variance of the posterior distribution. By setting $\eta = 0$ we obtain deterministic sampling equivalent to DDIM with discrete diffusion noise schedules and equivalent to flow-based methods (Lipman et al., 2023) when solving the generative ODE with the Euler method. However, by setting $\eta = 1$ we enable stochastic sampling with general noise schedules (Gaussian probability paths in the context of flow matching), which has not yet been explored up to our best knowledge.

## B. Graph-Sequential sampling

The graph-based sampling order is motivated by the idea that the dependency structure between patches can be exploited to estimate the level of knowledge about a given patch $i$ based on the noise levels of patches connected to it. By leveraging these dependencies, the sampling process can be guided in a structured and efficient manner, ensuring that patches with stronger conditioning from their neighbours are prioritized during denoising.

The adjacency matrix is defined as $A \in \mathbb{R}^{(N \cdot M) \times (N \cdot M)}$, where $(N, M)$ are the dimensions of the image patch grid. In the case of Sudoku, this grid is of size $(9, 9)$. The adjacency matrix is a binary mask where all elements of the same row, column, or $3 \times 3$ subgrid are connected. Other elements are not connected (represented by a value of $0$ in the adjacency matrix).

When selecting the next patch to evaluate, we take $\arg\max_{patch}((\mathbf{1} - t) \cdot A) \odot K$, where $\mathbf{t} \in [0, 1]^{M \cdot N}$ is the current noise level vector, and $K \in \{0, 1\}$ is a mask with $1$ for all patches that didn't start denoising process and $0$ otherwise. So we propagate $\mathbf{1} - \mathbf{t}$ (denoising level) through the graph, and select a patch that has the noise level $t_i = 1$, but has the strongest conditioning from its predecessors.

## B.1. Drawbacks

The certainty obtained using this algorithm is solely noise-level based. It cannot distinguish whether a patch is being conditioned by the same number across its row, column, and subgrid (weak conditioning) or by three different numbers (strong conditioning). As a result, the model with predicted uncertainty (as described in Sec. 3.2.2) can surpass the performance of the graph-based approach (Tab. 1).

## C. Uniform $\bar{t}$ sampling

For better intuition of why Uniform $\bar{t}$ should be one of our goals, we show the distributions of individual $t_i$ and $\bar{t}$ in Fig. 9.

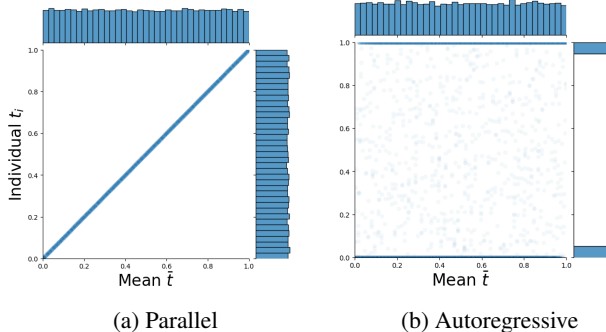

(a) Parallel          (b) Autoregressive

Figure 9: The distributions of individual patch noise level $t_i$ and mean over the image $\bar{t}$ in parallel sampling (left) and autoregressive generation (right). In a parallel generation, all patches have the same noise level throughout inference, so both $t_i$ and $\bar{t}$ marginal distributions are uniform. Autoregressive generation's $t_i$ distribution is very dense around 0 and 1, as during inference only one patch can have value other than 0 or 1 – the one currently generated. It's important, that regardless of which inference strategy we choose, the distribution of $\bar{t}$ remains uniform.

### C.1. Recursive allocation sampling $\mathbf{t} \sim p(\mathbf{t} \mid \bar{t})$

To sample $\mathbf{t} \sim p(\mathbf{t} \mid \bar{t})$, where $\mathbf{t} \in [0,1]^d$, we generate a vector with a specified mean $\bar{t}$ by a recursive sum allocation algorithm. The key idea is to define the total sum of the sampled vector as $s = d \cdot \bar{t}$, then recursively partition $s$ into two sum contributions from the first and second halves of the vector. This process is illustrated in Algorithm 1.

---

**Algorithm 1** Recursive Sampling of a sum constrained Vector

---

**Require:** $s$ (total sum), $d$ (vector dimension)
**Ensure:** A vector $\mathbf{x} \in [0,1]^d$ such that $\sum x_i = s$
  1: **Function** GetSumConstrainedVector($S, d$)
  2: **if** $d = 1$ **then**
  3:    **output** $[S]$ {End of recursion for 1 element}
  4: **end if**
  5: $d_1 \leftarrow \lfloor d/2 \rfloor$ {Get Split dimension}
  6: $d_2 \leftarrow d - d_1$
  7: $s_1^{\max} \leftarrow \min(s, d_1)$ {Define upper bounds of the contributions}
  8: $s_2^{\max} \leftarrow \min(s, d_2)$
  9: $s_1^{\min} \leftarrow \max(0, s - s_2^{\max})$ {Define lower bound}
10: Sample $r \sim p_{\text{split}}(r|d)$, where $r \in [0,1]$ {Sample a split point}
11: $s_1 \leftarrow s_1^{\min} + (s_1^{\max} - s_1^{\min}) \cdot r$
12: $s_2 \leftarrow s - s_1$
13: **output** GetSumConstrainedVector($s_1, d_1$) $\cup$ GetSumConstrainedVector($s_2, d_2$)

---

The algorithm first determines the range of possible contributions from the first half of the vector, $(s_1^{min}, s_1^{max})$. Then it samples a split point from a distribution $p_{split}(r|d)$, which has support on $[0,1]$. The sampled value is then scaled to the appropriate range of split points.

The split distribution $p_{split}(d)$ is modelled as a symmetrical case of a Beta distribution, empirically tuned to match the natural mid-point splitting behaviour of independently sampled uniform noise vectors. Explicitly,

$$p_{split}(r|d) = Beta(r|\alpha, \beta), \tag{25}$$

where

$$\alpha = \beta = (d - 1 - (d \bmod 2))^{1.05} \cdot sharpness. \tag{26}$$

In a uniformly sampled vector $\mathbf{t} \sim \mathcal{U}([0,1]^n)$ as the number of patches $d$ increases, extreme split points become less likely. For example with 100 patches and $\bar{t} = 0.5$ $s_1$ could be between 0 and 50 but it is highly unlikely that all patches in the first half of the vector will be ones and all in the second half will be zeros. Instead, $s_1$ is most likely to be close to 25. That is why the split point distribution gets more center-heavy with higher vector dimensionality $d$ – we model this by increasing the $\alpha$ and $\beta$ parameters for higher $d$ as in Eq. 26. For $sharpness = 1$ and $\bar{t} = 0.5$ the distribution of $\mathbf{t} \in [0,1]^n$ closely resembles $\mathcal{U}([0,1]^n)$ regardless of $n$. Higher $sharpness$ results in distributions more tightly centred around $\bar{t}$, while $sharpness \in (0,1)$ leads to oversampling values closer to 0 and 1. The effect of the $sharpness$ parameter on the distribution is illustrated in Fig. 10. Its influence on the Even Pixels dataset is presented in Tab. 7.

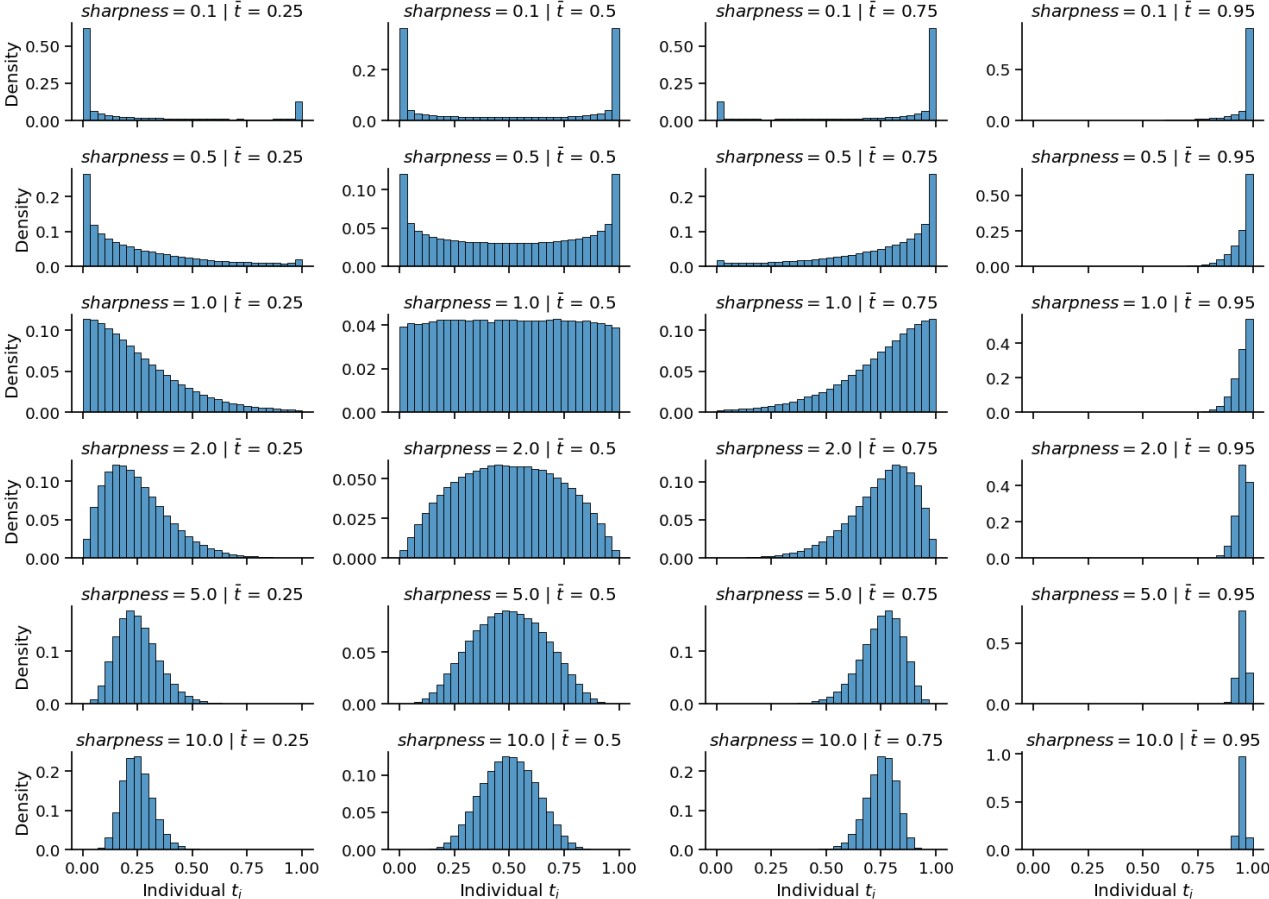

Figure 10: Distributions of $t_i$ noise level values given different *sharpness* ($x$-axis) and mean noise level $\bar{t}$.

The advantage of this algorithm over, for example, iterative perturbation methods is that it can be easily parallelized across the batch dimension, ensuring efficient utilization of computational resources. Additionally, it exhibits stable execution time, a crucial property for maintaining predictable and consistent performance in the training pipeline.

# D. Even Pixels Evaluation

For evaluation, we generate a hue histogram from the model's output and identify the peak hue $h_{\max}$. We then count the number of pixels closer to $h_{\max}$ than to its complementary hue $h^*_{\max} = (h_{\max} + 180°) \mod 360°$. The **error** is computed as the absolute difference between this count and half the total number of pixels

$$\mathbf{error} = \left| \#(\text{pixels closer to } h_{\max} \text{ than } h^*_{\max}) - \frac{\text{width} \cdot \text{height}}{2} \right|$$

The **accuracy** indicates in which fraction of samples where the pixel hue counts equal (**error** $= 0$).

# E. Counting Polygons/Stars Dataset

### E.1. objects color selection

The color of the numbers and polygons/stars is chosen in *HSV* space $\text{hsv}(c, 1.0, 0.9)$, where

$$c = \arg \min_h \left\{ (\text{histogram}(H_{img}) * G_\sigma)(h) \right\}$$

and $G_\sigma$ is a Gaussian kernel with a small $\sigma$. In other words, we search for the hue with the lowest count in the smoothed hue histogram of an image. We found this method as an intuitive approach to increase the visibility of the objects on the FFHQ background.

### E.2. Evaluation

To assess the generative model's ability to learn the implicit counting constraint, for each of the object types (polygons or stars) we employ a ResNet-50-based classifier with three prediction heads:

- Predicting the set of two numbers $(N, K)$ present in the image.

- Predicting the number of polygons/stars.

- Predicting the number of object vertices.

For star objects, we only count the convex, outward-facing vertices. The classifiers are trained on a dataset with similar image compositions, but the consistency constraint between the numbers and the actual object counts is dropped in $50\%$ of cases. For all the prediction tasks mentioned above, both classifiers have achieved over $99.9\%$ accuracy in the validation set.

For evaluation, the classifier detects the numbers present in the generated image and compares them to the true object and vertex/spikes counts. A generated image is classified as correct if the set of extracted numbers is the same as the set of actual polygon/star objects and vertex counts (regardless of order): correct $\Leftrightarrow \{\text{num}_1, \text{num}_2\} = \{\#\text{objects}, \#\text{vertices}\}$.

The evaluation metric of the generative model, **accuracy**, is computed as the fraction of images for which the number-to-object-count correspondence is correct.

# F. Implementation Details

For reasoning in the spatial visual domain, we divide images into patches of a fixed size. While we use the size of a cell, i.e., MNIST example $28 \times 28$ as the patch size for the MNIST Sudoku dataset with image resolution $252 \times 252$, we choose patch sizes of $4 \times 4$ and $8 \times 8$ for the even pixel and counting polygons/stars datasets with resolutions $32 \times 32$ and $128 \times 128$, respectively.

Although SRMs are agnostic w.r.t. the architecture choice, we choose simple 2D UNets for all experiments, as they are widely established for image generation. We provide all architecture and training hyperparameters in Tab. 5.

For training the SRMs with individual noise levels per patch and the estimation of its uncertainty in the noise prediction, we additionally apply to simple architecture modifications. Instead of conditioning the UNet on a single encoding of the noise level $t$, we compute a map with per-pixel noise levels, where all pixels within a patch share the same $t$. This map is

| | MNIST Sudoku | Even Pixels | Counting Polygons/Stars FFHQ |
|---|---|---|---|
| Channels | 128 | 64 | 128 |
| Depth | 2 | 2 | 2 |
| Channel multipliers | 1, 1, 2, 2, 4, 4 | 1, 2, 2, 4 | 1, 1, 2, 2, 4 |
| Head channels | 64 | 64 | 64 |
| Attention resolution | 16, 8 | 8, 4 | 16, 8 |
| Parameters | 118M | 19.7M | 76.8M |
| Effective batch size | 56 | 2048 | 56 |
| Iterations | 250k | 100k | 250k |
| Learning Rate | 1e-4 | 8e-4 | 1e-4 |

Table 5: **Hyperparameters used for all experiments.**

| | Deterministic | | Stochastic | |
|---|---|---|---|---|
| Sampling | L1↓ | Acc↑ | L1↓ | Acc↑ |
| Parallel | 26.288 | 0.004 | 19.156 | 0.010 |
| Predicted Order w/o Overlap | 3.652 | 0.500 | **3.212** | **0.516** |
| Predicted Order + Overlap | 4.192 | 0.420 | 4.340 | 0.422 |
| Graph-based Order w/o Overlap | 6.212 | 0.240 | 5.816 | 0.266 |
| Graph-based Order + Overlap | 6.652 | 0.188 | 6.036 | 0.238 |

Table 6: **Ablation over Deterministic vs. Stochastic Sampling on MNIST Sudoku hard.** Stochastic sampling almost consistently outperforms deterministic sampling, highlighting the benefits of our generative process combining arbitrary (non-diffusion) noise schedules with stochastic sampling.

| Sharpness | 0.50 | | 0.75 | | 1.00 | |
|---|---|---|---|---|---|---|
| Sampling | #E↓ | Acc↑ | #E↓ | Acc↑ | #E↓ | Acc↑ |
| Parallel | 3.580 | 0.096 | 5.184 | 0.054 | 4.076 | 0.074 |
| Predicted Order + Overlap | 0.840 | 0.368 | **0.534** | **0.518** | 0.590 | 0.484 |
| Random Order + Overlap | 0.776 | 0.342 | 0.584 | 0.476 | 0.656 | 0.430 |

Table 7: **Sharpness Ablation on Even Pixels.** By lowering the sharpness hyperparameter of our noise level sampling algorithm, we can train SRMs to be more tailored towards sequential sampling methods.

bilinearly interpolated to the spatial resolution of the feature maps in each layer, encoded using a shared MLP, and finally used to compute scale and shift maps applied pixel-wise to the features. While this modification is specific to UNets, similar adjustments can be made for other architectures like transformers, which already consider patches as tokens in the vision domain. In order to predict a single uncertainty per patch, our denoising network outputs an additional one-dimensional feature map, on which we apply 2D average pooling with the kernel size and stride corresponding to the patch size of our spatial variables. The result is interpreted as log-variance of the predicted noise.

# G. Additional Ablations

We provide additional ablation studies considering hyperparameter choices for our experiments. Tab. 6 compares the performance of deterministic and stochastic sampling on the hard difficulty of the MNIST Sudoku dataset by setting either $\eta = 0$ or $\eta = 1$ in the generation process described in Sec. A. We can see that stochastic sampling is favorable for spatial reasoning, which we enable in combination with arbitrary noise schedules.

In Tab. 7, we ablate the choice of the sharpness hyperparameter in our noise level sampling algorithm for training SRMs. On the Even Pixels dataset, we can oversample noise level combinations for spatial variables that are more likely in sequential sampling strategies by lowering the sharpness. This results in higher performance depending on the data distribution.

| Metric | Sampling | Easy | | Medium | | Hard | |
| | | UNet | DiT | UNet | DiT | UNet | DiT |
|---|---|---|---|---|---|---|---|
| Accuracy ↑ | Parallel | 0.998 | 0.992 | 0.590 | 0.328 | 0.010 | 0.000 |
| | Predicted Order w/o Overlap | 0.998 | 0.988 | **0.754** | **0.618** | **0.516** | **0.248** |
| | Predicted Order + Overlap | **1.000** | **0.992** | 0.716 | 0.598 | 0.422 | 0.166 |
| | Random Order w/o Overlap | 0.998 | 0.992 | 0.568 | 0.346 | 0.024 | 0.004 |
| | Random Order + Overlap | **1.000** | 0.974 | 0.542 | 0.330 | 0.020 | 0.004 |
| | Graph-based Order w/o Overlap | 0.996 | 0.978 | 0.662 | 0.470 | 0.266 | 0.086 |
| | Graph-based Order + Overlap | **1.000** | **0.992** | 0.576 | 0.430 | 0.238 | 0.056 |
| L1 Error ↓ | Parallel | 0.012 | 0.044 | 3.312 | 10.940 | 19.156 | 39.048 |
| | Predicted Order w/o Overlap | 0.012 | 0.068 | **1.616** | **3.960** | **3.212** | **8.620** |
| | Predicted Order + Overlap | **0.000** | 0.060 | 1.816 | 4.216 | 4.340 | 11.244 |
| | Random Order w/o Overlap | 0.012 | 0.052 | 3.828 | 9.184 | 12.612 | 23.792 |
| | Random Order + Overlap | **0.000** | 0.172 | 4.172 | 10.324 | 13.896 | 25.368 |
| | Graph-based Order w/o Overlap | 0.020 | 0.180 | 2.544 | 5.864 | 5.816 | 12.752 |
| | Graph-based Order + Overlap | **0.000** | 0.072 | 3.240 | 7.100 | 6.036 | 14.912 |

Table 8: **Denoising Architecture Ablation on MNIST Sudoku.** We compare SRM variantes using UNet and DiT denoising backbones. For each metric, model, and difficulty, the best-performing configuration is highlighted in bold. Sequential generation with predicted order yields the strongest performance across both metrics independent of the used architecture.

| Metric | Sampling | Easy | | Medium | | Hard | |
| | | Linear | Cosine | Linear | Cosine | Linear | Cosine |
|---|---|---|---|---|---|---|---|
| Accuracy ↑ | Parallel | 0.998 | **1.000** | 0.590 | **0.622** | 0.010 | **0.018** |
| | Predicted Order w/o Overlap | 0.998 | **1.000** | **0.754** | 0.748 | **0.516** | 0.378 |
| | Predicted Order + Overlap | **1.000** | 0.998 | **0.716** | 0.706 | **0.422** | 0.298 |
| | Random Order w/o Overlap | 0.998 | **1.000** | **0.568** | 0.502 | 0.024 | **0.036** |
| | Random Order + Overlap | **1.000** | 0.994 | **0.542** | 0.510 | 0.020 | **0.024** |
| | Graph-based Order w/o Overlap | 0.996 | **1.000** | **0.662** | 0.630 | **0.266** | 0.220 |
| | Graph-based Order + Overlap | **1.000** | 0.996 | 0.576 | **0.620** | **0.238** | 0.210 |
| L1 Error ↓ | Parallel | 0.012 | **0.000** | 3.312 | **3.048** | 19.156 | **17.140** |
| | Predicted Order w/o Overlap | 0.012 | **0.000** | **1.616** | 1.812 | **3.212** | 5.140 |
| | Predicted Order + Overlap | **0.000** | 0.012 | **1.816** | 2.056 | **4.340** | 5.920 |
| | Random Order w/o Overlap | 0.012 | **0.000** | **3.828** | 4.812 | **12.612** | 13.232 |
| | Random Order + Overlap | **0.000** | 0.032 | **4.172** | 5.020 | **13.896** | 14.200 |
| | Graph-based Order w/o Overlap | 0.020 | **0.000** | **2.544** | 2.752 | **5.816** | 6.592 |
| | Graph-based Order + Overlap | **0.000** | 0.032 | 3.240 | **2.924** | **6.036** | 6.848 |

Table 9: **Noise Level Schedule Ablation on MNIST Sudoku.** We compare SRM variants using Linear (Liu et al., 2023) and Cosine (Nichol & Dhariwal, 2021) noise level schedules. The winning schedule is highlighted in bold for each row. While the linear schedule clearly outperforms the cosine one for our setting, the benefits of sequentialization with a meaningful order remain independent of the choice of the noise level schedule.

To further demonstrate the architecture-agnostic nature of the SRM framework, we also trained a Diffusion Transformer (DiT B) (Peebles & Xie, 2023) with a patch size of 7 and 130M parameters on our MNIST Sudoku benchmark. The results, compared to those of a 2D UNet are presented in Tab. 8. Notably, in this highly structured domain, sequential generation with predicted variable order yields the best performance ndependent of the used architecture.

As an alternative to the linear schedule used in Rectified Flows (Liu et al., 2023), we also explored the cosine noise level schedule (Nichol & Dhariwal, 2021). The results are presented in Tab. 9. Independent of the noise level schedule, sequentialization with predicted order consistently outperforms all other sampling strategies. Interestingly, the linear schedule

seems to be more suitable than the cosine one for the dataset at hand. We suspect that this is because the cosine schedule allocates more time steps to low-noise regions, effectively focusing the denoising process on the high-frequency details, which are not relevant for the digit identity and thus not captured by our metrics.

## H. Sudoku Completion with LLMs

To evaluate the difficulty of generation-based Sudoku solving, we examined the performance of general-purpose large language models (LLMs). Specifically, we tested GPT-4o, GPT-4o-mini, and open-source Phi-4 on a text-based Sudoku completion task. Each model was given a textual representation of a Sudoku grid with missing elements and was tasked with filling in the blanks to complete the puzzle. The prompts included the full context of the Sudoku task: a description of the rules, output format, and three few-shot examples. An example is shown below:

```
[Example 1]
Input:
<sudoku>
34_|179|258
187|523|964
529|648|371
---+---+---
965|832|417
472|_16|835
_13|754|629
---+---+---
798|261|_43
631|485|792
_54|397|18_
</sudoku>

Output:
<sudoku>
346|179|258
187|523|964
529|648|371
---+---+---
965|832|417
472|916|835
813|754|629
---+---+---
798|261|543
631|485|792
254|397|186
</sudoku>
```

If a model's output deviated from the required format, we resampled until a well-formed 9×9 grid of digits — suitable for evaluation — was produced. The model was then asked to provide only the full solution, with no "inner thoughts". This yielded an autoregressive generation variant similar to our image-based generation approach, but in the text domain.

Table 10 compares the performance of the image-based Diffusion and SRM models with the few-shot LLMs across different difficulty levels. Although this comparison is not fair – our models are trained specifically for the task, in image space, whereas LLMs are few-shot prompted via text – it highlights the difficulty of solving Sudoku via generative models.

Figure 11 further breaks down model performance into validity, consistency, and overall accuracy as the number of masked cells increases. Validity measures whether the output follows the Sudoku rules, consistency ensures that the solution matches the initial grid, and accuracy requires both. Interestingly, while performance drops with increasing difficulty, accuracy improves when most of the board is masked, possibly due to resemblance with the few-shot examples in the prompt.

| Method | Modality | Setup | Sudoku Difficulty | | |
| --- | --- | --- | --- | --- | --- |
| | | | Easy | Medium | Hard |
| Diffusion | Image | Trained | 0.994 | 0.536 | 0.008 |
| **SRM (Ours)** | | | **0.998** | **0.754** | **0.516** |
| Phi-4 | Text | Few-shot | 0.038 | 0.000 | 0.000 |
| GPT-4o-mini | | | 0.205 | 0.000 | 0.000 |
| GPT-4o | | | 0.556 | 0.001 | 0.011 |

Table 10: **SRM vs LLM.** To highlight the complexity of our MNIST Sudoku benchmark, we compare Sudoku solving capabilities of image generators with general-purpose LLMs used in a few-shot setting. Note that this is not a fair comparison, as SRM and Diffusion are trained directly for the task. Despite tackling a discrete and therefore simpler version of Sudoku, LLMs perform poorly in this task compared to SRM.

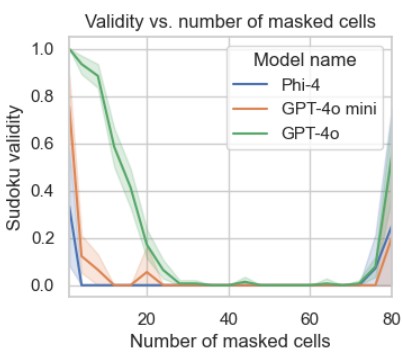
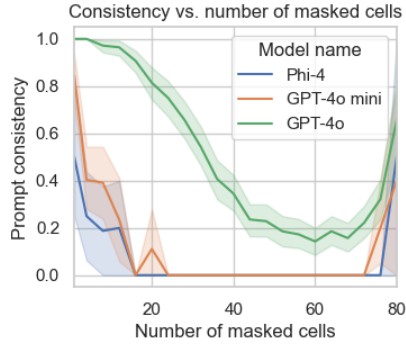
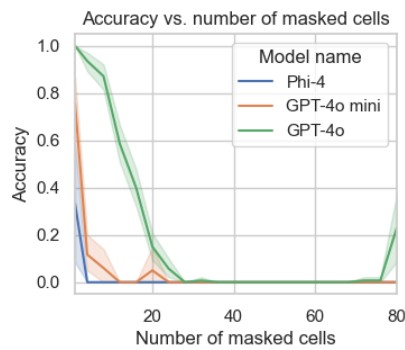

(a) Validity: proportion of outputs that are proper Sudoku grids (no duplicate digits).

(b) Consistency: solutions that match the initially masked grid entries.

(c) Accuracy: solutions that are both valid and consistent.

Figure 11: **LLM Performance in Discrete Sudoku Completion.** As the number of masked cells increases corresponding to an increasing difficulty level of the Sudoku completion task, both validity and consistency drop. Surprisingly, when almost all elements are masked, the accuracy improves – possibly due to similarity to some example grids in the prompt.

We additionally experimented with a chain-of-thought setup, where the model was prompted to iteratively fill one cell at a time, selecting its generation order. However, consistency with the input grid was insufficient for a reliable evaluation. We also tested the Deepseek-R1 reasoning model, allowing unconstrained internal reasoning before producing a final output. Although the model attempted to verify its own solutions, it frequently entered loops of self-correction or made mistakes while evaluating the constraints, resulting in unfinished rollouts or invalid Sudokus. Overall, the results suggest that solving Sudoku, even in simplified textual form, remains a challenging task for general-purpose generative models.

