# OpenReview forum: "Spatial Reasoning with Denoising Models"
_ICML.cc/2025/Conference — ICML 2025 poster_

### Official Review · Reviewer_FTjC · 2025-03-12

**Overall Recommendation:** 4

**Summary:**

The authors investigate the application of diffusion models as solvers of probabilistic inference over continuous variables, which accommodates various problems in spatial reasoning.  A key consideration is the various possible decompositions of the joint distribution of unobserved variables, and how some decompositions may be beneficial.  Another consideration is the noise schedule used traditionally in diffusion models, being a fixed increment, and the drawbacks of diffusion forcing leading to undertraining for early/late inference steps.  By tackling those two aspects in tandem, the authors propose to explore the design space spanning parallel generation to autoregressive generation, proposing a "recursive allocation sampling" algorithm with controllable sharpness, making it adaptable to different inference scenarios.  The benefits of the proposed scheme is demonstrated on a challenging MNIST-Sudoku benchmark, with 3 difficulty levels, as well as an even-coloring experiment.  Finally, the authors attempt to highlight more realistic settings by counting a small number of geometric primitives overlayed on backgrounds sampled from the FFHQ dataset, highlighting some of the remaining challenges.

**Claims And Evidence:**

Yes.  Remaining questions were adequately highlighted following the conclusions.

**Essential References Not Discussed:**

N/A

**Experimental Designs Or Analyses:**

Yes, read through Section 4.  Some implementation details and ablations are deferred to the appendices, which is common.

**Methods And Evaluation Criteria:**

Yes, albeit a bit primitive suitable for an early conceptual investigation as this one.

**Other Comments Or Suggestions:**

N/A

**Other Strengths And Weaknesses:**

- The FFHQ backgrounds is mentioned as a confusing factor for the "counting polygons" experiment.
- - It would have been nice to include an ablation with e.g. blank backgrounds to test this further
- - This is potentially significant as this experiment doesn't strongly support the claimed benefits of sample orderings

**Questions For Authors:**

No further questions at this time.

**Relation To Broader Scientific Literature:**

Reasoning over complex domains is certainly relevant in science, although this particular paper is a bit more primitive or too early to be of broader relevance.

**Theoretical Claims:**

The derivations in Appendix A seemed correct, although it wasn't immediately clear which "statement/result" the "Proof" aims to establish.
- Please clarify, marking a clear Proposition, Lemma, or Theorem.

---

> ### Author Rebuttal · Authors · 2025-04-01
>
> We thank you very much for your time and positive feedback. We are happy to see that you agree with us that reasoning over complex domains is certainly relevant in science.
>
> > Missing Proposition, Lemma, or Theorem in Appendix A
>
> Thank you for pointing out this lack of clarity. On a high level, the Appendix A shows that the DDIM \[1\] formulation can be extended to noise schedules introduced in the context of flow matching \[2\]. By considering Gaussian reverse distributions as in DDPM / DDIM instead of ODEs and SDEs with flow-based models, we explicitly show that improvements of diffusion models such as a learned variance \[3\] can be combined with non-diffusion noise schedules like from rectified flows \[2\]. To the best of our knowledge, there have been no flow-based formulations incorporating a learned variance so far. To answer the question about the individual proofs:
>
> 1. The first proof shows that the chosen mean of the Gaussian reverse distribution for the next (less noisy) state $x\_{t\_{i-1}}$ given the current noisy state $x\_{t\_i}$ and the clean data $x\_0$ ensures that the marginal distribution only conditioned on the clean data $x\_0$ is of the desired form with the noise schedule $a, b$ defining the interpolation weights of data and noise, respectively.
> 2. Since the standard deviation in the reverse distribution is a free variable, we follow DDIM by defining it as an interpolation between deterministic sampling (zero standard deviation) and stochastic sampling with a Markovian forward process, which makes DDIM equivalent to DDPM. Analogously, the second proof validates that the specific choice of the standard deviation in the reverse distribution results in a Markovian forward process, but for our formulation with more general noise schedules.
>
> As you suggested, we will mark these statements as clear Propositions in the final version of the paper.
>
> - \[1\] Denoising Diffusion Implicit Models. ICLR 2021
> - \[2\] Flow Matching for Generative Modeling. ICLR 2023
> - \[3\] Improved Denoising Diffusion Probabilistic Models. ICLR 2021

---

### Official Review · Reviewer_KdD5 · 2025-03-14

**Overall Recommendation:** 2

**Summary:**

This paper introduces “Spatial Reasoning Models” (SRMs), a framework for performing high-level reasoning across sets of continuous variables in diffusion/flow-based generative models. By allowing each spatial variable (e.g., an image patch) to have its own noise level, SRMs can systematically add or remove noise in different regions of an image in a sequential or partially sequential manner. This approach is shown to reduce “hallucinations” common to standard diffusion models on tasks such as Sudoku with MNIST digits, balancing pixel colors, or matching the digit count of polygons

## update after rebuttal

**Claims And Evidence:**

- In line 190, the paper suggests that the mean noise level ( \bar{t} ) should ideally follow a uniform distribution with a sufficient number of inference steps. Is there a rigorous theoritical justification for it?

**Essential References Not Discussed:**

No

**Experimental Designs Or Analyses:**

Yes

**Methods And Evaluation Criteria:**

- What's the role of Appendix A? If it is about the diffusion process for the single (possibly higher-dimensional) continuous random variable,   the concrete formulation and proofs have already been well-defined in previous works.
- The difference between the proposal and spatially autoregressive approaches is really confusing. If I understand correctly, when the proposed algorithm utilize random, predicted uncertainty, or manual graph-based order of sequentialization, it still generates each patch one by one.

**Other Comments Or Suggestions:**

No

**Other Strengths And Weaknesses:**

No

**Questions For Authors:**

No

**Relation To Broader Scientific Literature:**

Reasoning in diffuison generation in high-depent multi-variables.

**Theoretical Claims:**

- I think the paper does not provide any meaningful theoritical claims.

---

> ### Author Rebuttal · Authors · 2025-04-01
>
> Thank you very much for your review. We address your concerns as follows:
>
> > Justification for uniform distribution of mean noise level during training
>
> 1. We would like to refer you to Fig. 8 of our paper’s Appendix. It shows that for the two extreme cases of parallel and autoregressive generation, the observed mean noise levels during inference form a uniform distribution. For all cases in between these, like our sequential sampling with overlap, it therefore also makes sense to train the model exactly for this mean noise level distribution. However, since this is an intuitive rather than rigorous theoretical justification, we are willing to reformulate this precisely in the paper to avoid using the term “ideally”.
> 2. Moreover, previous works like StableDiffusion3 have investigated different noise level weightings such as a logit normal distribution to oversample intermediate noise levels, which can significantly improve performance in certain tasks like image generation. Our two-step noise level sampling strategy during training is directly compatible with such weightings by replacing the uniform distribution for the mean with the logit normal distribution, for example. We will make sure to clarify the flexibility w.r.t. the choice of distribution for the mean noise level in the final version and would like to further investigate this direction in future work.
>
> > Role of Appendix A
>
> Thank you for pointing out this lack of clarity.
>
> 1. We agree that many previous works such as \[1, 2, 3\] introduce the concrete formulations of diffusion, flow matching, or unifying frameworks. Our Appendix A does not reinvent the wheel, but shows that the DDIM \[1\] formulation can be extended to noise schedules introduced in the context of flow matching. By considering Gaussian reverse distributions as in DDPM / DDIM instead of ODEs and SDEs with flow-based models, we explicitly show that improvements of diffusion models such as a learned variance \[4\] can be combined with non-diffusion noise schedules like from rectified flows. To the best of our knowledge, there have been no flow-based formulations incorporating a learned variance so far. We will clearly specify the role of Appendix A in the final version.
>
> - \[1\] Denoising Diffusion Implicit Models. ICLR 2021
> - \[2\] Flow Matching for Generative Modeling. ICLR 2023
> - \[3\] Stochastic Interpolants: A Unifying Framework for Flows and Diffusions. arxiv 2023
> - \[4\] Improved Denoising Diffusion Probabilistic Models. ICLR 2021
>
> 2. Please note that we have chosen not to claim this as one of our main contributions (cf. contributions and key findings in introduction), as we are aware of the many prior works introducing the theoretical formulations of denoising generative models. Our SRM framework can benefit from any improvements of the diffusion process for single continuous random variables, as our formulation of spatial reasoning is orthogonal to that.
>
> > Confusion w.r.t. spatially autoregressive approaches
>
> You are correct that sequentialization **without overlap**, utilizing random, predicted uncertainty or manual graph-based order, results in spatially autoregressive patch-by-patch generation. Our framework enables the training of one model that can be used in combination with multiple different sampling strategies including parallel generation (synchronous denoising of all variables) as with usual diffusion models, spatially autoregressive generation, and everything in between, e.g., sequentialization **with overlap**. In addition to the amount of sequentialization as one investigated degree of freedom (cf. Fig. 3), we propose and evaluate different orders, in which variables (patches) are chosen for denoising, with the uncertainty- and graph-based orders being novel w.r.t. prior works for denoising-based spatially autoregressive generation. Our paper shows that, for the same trained model, different sampling strategies significantly impact the level of hallucination and that the best strategy depends on the data distribution, e.g., non-overlapping sequentialization for MNIST Sudoku and a high overlap for Even Pixels.
> We will make sure to clarify this in the final version.

---

### Official Review · Reviewer_G4qT · 2025-03-17

**Overall Recommendation:** 4

**Summary:**

This paper introduces Spatial Reasoning Models (SRMs), a framework for performing reasoning over sets of continuous variables using denoising generative models. The authors observe that standard diffusion/flow models often collapse to hallucination when handling complex distributions. The key innovations include: enabling different noise levels for different spatial variables, a novel "Uniform $\bar{t}$" noise sampling strategy that better aligns training and inference distributions, uncertainty estimation to guide the generation process, and various sequentialization strategies.

The paper introduces three benchmark tasks to evaluate reasoning capabilities: (1) MNIST Sudoku, where models must complete partially filled Sudoku grids using MNIST digits; (2) Counting Pixels, requiring balanced color distribution; and (3) Counting Polygons, testing understanding of relationships between numbers and visual elements. Through extensive experimentation, the authors demonstrate that sequentialization in generation and uncertainty-based ordering significantly improves reasoning capabilities, increasing accuracy from <1% to >50% on hard reasoning tasks.

**Claims And Evidence:**

Most of the claims made are supported by various empirical evidence through different experimets on the introduced benchmarks. However, there are few claims that lack adequate evidence or contain some inconsistencies.

1. While the paper shows improvement on all three benchmarks, the performance on the most realistic task (Counting Polygons FFHQ) is modest (18.6% vs 13.2% baseline). The paper doesn't sufficiently demonstrate that these improvements would translate to more complex real-world reasoning scenarios.

2. The paper claims their formulation works with various noise schedules (DDPM, rectified flows, etc.) in Section 3.1.1, but doesn't provide comparative results across different schedules to validate this.

3. In Section 4.2, the authors state "While SRMs are agnostic to different architectures," but provide experiments only with 2D UNets.

4. For the Counting Polygons FFHQ evaluation, the authors rely on a ResNet classifier to determine correctness, but don't report this classifier's accuracy, making it impossible to assess the reliability of the reported metrics. Since the authors don't report the accuracy of this ResNet classifier itself, we don't know how reliable the evaluation is. If the classifier makes errors in detecting numbers or counting polygons/vertices, the reported model performance metrics ( for example reported numbers like 18.6%, 13.2%) would be affected.

5. The paper doesn't adequately explain why sequentialization yields dramatic improvements on Sudoku (>50%) but only modest gains on Counting Polygons (~5%).

**Essential References Not Discussed:**

NA

**Experimental Designs Or Analyses:**

I examined the experimental designs for the three benchmarks (MNIST Sudoku, Even Pixels, Counting Polygons) and ablation studies. The Counting Polygons experiment relies on a separately trained classifier for evaluation, but the paper doesn't report this classifier's accuracy, introducing potential measurement bias. This issue is discussed earlier.

**Methods And Evaluation Criteria:**

Yes

**Other Comments Or Suggestions:**

- Line 603: VLB should be written in the full form before being used as short form
- Figure 7: the x-axis label "Overlap" would be clearer if it specified "Overlap Ratio" or something similar.
- The term "sharpness" is used in Section 3.2.1 without proper definition until much later in the appendix.

**Other Strengths And Weaknesses:**

**Strengths**

**S1. Originality**: The paper introduces several novel ideas including different noise levels for different spatial variables, a "Uniform $\bar{t}$" noise sampling strategy that better aligns training and inference distributions, uncertainty estimation to guide generation, and various sequentialization strategies.

**S2. Benchmarking**: The proposed benchmarks provide a systematic way to evaluate and quantify reasoning capabilities and hallucination in generative models.

**S3. Experimentation and Performance**: The paper demonstrates significant improvements over baseline diffusion models (from <1% to >50% accuracy on hard Sudoku), highlighting the effectiveness of the proposed approach for spatial reasoning tasks.



**Weaknesses**

**W1. Real-world applicability**: While the paper shows improvement on all three benchmarks, the performance on the most realistic task (Counting Polygons FFHQ) is modest (18.6% vs 13.2% baseline). The paper doesn't sufficiently demonstrate that these improvements would translate to more complex real-world reasoning scenarios.

**W2. Limited validation of noise schedules**: The paper claims their formulation works with various noise schedules (DDPM, rectified flows, etc.) in Section 3.1.1, but doesn't provide comparative results across different schedules to validate this.

**W3. Architecture limitations**: In Section 4.2, the authors state "While SRMs are agnostic to different architectures," but provide experiments only with 2D UNets.

**W4. ResNet Accuracy**: For the Counting Polygons FFHQ evaluation, the authors rely on a ResNet classifier to determine correctness without reporting this classifier's accuracy, making it impossible to assess the reliability of the reported metrics (e.g., 18.6%, 13.2%).

**W5. Inconsistent improvements**: The paper doesn't properly explain why sequentialization yields dramatic improvements on Sudoku (>50%) but only modest gains on Counting Polygons (~5%).

**W6. Computational overhead**: There is a significant computational cost associated with sequential generation compared to parallel approaches, which the authors should have addressed more thoroughly.

**W7. Data Statistics**: Details about the number of data points used in evaluating each benchmark is reported. Authors should consider adding some detail about them.

**Questions For Authors:**

Q1: For the Counting Polygons FFHQ benchmark, what is the accuracy of the ResNet classifier used for evaluation?

Q2: What explains the dramatic improvement on Sudoku (\~ 50% gain) compared to the modest gains on Counting Polygons (~5%)? Is this due to fundamental differences in the tasks, model capacity limitations, or other factors?

Q3: Section 3.1.1 claims compatibility with various noise schedules (DDPM, rectified flows, etc.). Have you empirically verified performance across different schedules?

Q4: Could you quantify the computational cost difference between the parallel baseline and your best sequential approaches (e.g., inference time for Sudoku)?

I am open to increasing my score after satisfactory answers to all my questions and previously raised concerns.

**Relation To Broader Scientific Literature:**

The paper extends chain-of-thought reasoning concepts from language models to continuous spatial domains, while building upon sequential generation techniques from AR-Diffusion and Diffusion Forcing.The paper adequately covers all the relevant work.

**Theoretical Claims:**

I checked the theoretical claims and proofs throughout the paper and its appendices, including the mathematical formulation in Section 3, Appendix A and appendix C. I did not find any notable mathematical errors in the theoretical claims and proofs throughout the paper. However, I do not consider myself as an expert in the domain, so I might have overlooked some detail.

---

> ### Author Rebuttal · Authors · 2025-04-01
>
> We sincerely appreciate your constructive comments and are happy to see that you value the *originality* of our *novel ideas* achieving *significant improvements [...] highlighting the effectiveness of the proposed approach for spatial reasoning tasks*.
> > W1. Real-world applicability
>
> We agree that there is a gap between our benchmark and real-world reasoning scenarios. We designed our datasets to measure hallucination of denoising generative models, which is extremely difficult for real-world applications and already non-trivial for our counting polygons dataset. We share Reviewer FTjC’s opinion about it being “suitable for an early conceptual investigation as this one” and would like to investigate real-world applications in future work. Our framework enables everybody to perform research on a wide range of domains.
> > W2 + Q3. Validation of noise schedules
>
> We provide results for models trained on MNIST Sudoku with the cosine noise schedule commonly used in diffusion or flow-based models: [https://figshare.com/s/8b72019bc22a66bce0c8](https://figshare.com/s/8b72019bc22a66bce0c8)
> All our conclusions regarding the benefits of sequentialization with a meaningful order hold for the cosine schedule too. We add this ablation to the final version.
> > W3. Architecture limitations
>
> We provide results for models using a diffusion transformer (DiT B) with patch size 7 and 130M parameters, which roughly corresponds to the size of our 2D UNet: [https://figshare.com/s/2d87cf6657c1a948347d](https://figshare.com/s/2d87cf6657c1a948347d)
> All our conclusions hold for the DiT architecture too. However, we see generally worse performance than with the UNet, which we attribute to DiTs being less suited for denoising in pixel space. We will add this ablation to the final version.
> > W4 + Q1. ResNet Accuracy
>
> The classifier used for the Counting Polygons (and Stars, see next paragraph) evaluation has an accuracy of \>99.9% on a validation split such that reported metrics are reliable.
> > W5 + Q2. Inconsistent improvements
>
> 1. We attribute this to fundamental differences in the tasks. For MNST Sudoku, all numbers have the same sizes such that, during sampling with a parallel denoising strategy, the commitment to individual digits has to happen at similar points in time. Due to the spatial dependencies of Sudoku, this is suboptimal and a spatially autoregressive strategy together with a good order can commit to the digits in a cell-by-cell fashion.
>    For the Counting Polygons dataset, the numbers are high-frequency details compared to larger polygons. As a result, a coarse-to-fine generation with the diffusion baseline can first commit to a number of polygons of a certain vertex count and then generate matching digits.
>
> 2. To further validate our hypothesis, we conducted additional experiments on a modified benchmark version “Counting Stars”, for which we replace polygons with stars (cf. [https://figshare.com/s/6d8b59fd96f56f05b571](https://figshare.com/s/6d8b59fd96f56f05b571)). The motivation is that stars are composed of higher frequencies, moving the “point of commitment” to numbers and stars closer together in time.
>
> | Sampling | Counting Accuracy | Star Consistency |
> | :---- | :---- | :---- |
> | Diffusion Model | 0.070 | 0.544 |
> | Ours, Parallel | 0.034 | 0.576 |
> | Ours, Predicted Order w/o Overlap | 0.076 | 0.844 |
> | Ours, Predicted Order + Overlap | **0.150** | 0.888 |
> | Ours, Random Order w/o Overlap | 0.080 | 0.872 |
> | Ours, Random Order + Overlap | 0.104 | **0.938** |
>
> For the diffusion baseline, the accuracy of matching numbers and stars decreases significantly compared to the version with polygons (7% vs 13.2%), while sequential sampling with overlap and predicted order maintains the same performance. More interestingly, for parallel generation, we noticed hallucinations of samples with stars having inconsistent numbers of points (cf. star consistency column). For sequential sampling, the model can replicate stars after the first one has been generated, whereas in parallel sampling, the decisions over the number of points for all stars are again closer in time. We hypothesize that this behavior was not visible for polygons because of differences in terms of frequencies, with the diffusion model having more “time to correct itself” for low-frequency polygons, as their generation starts earlier than for the high-frequency stars.
> > W6 + Q4. Computational cost
>
> In all our experiments, we set the total number of denoising steps to 1000\. This means that the computational cost is equal for all sampling methods resulting in a fair comparison. This also means that individual variables have fewer steps the “more sequential” the sampling is (lower overlap) but it still performs better. We will thoroughly describe this aspect in the final version.
> > W7. Data Statistics
>
> Thank you for pointing out this missing detail that we will add to the paper. For all evaluations, we sample 500 data points to compute metrics.

---

### Official Review · Reviewer_7X1h · 2025-03-22

**Overall Recommendation:** 2

**Summary:**

This paper studies how diffusion models perform on higher-level reasoning tasks, such as the Sudoku game. The authors introduce a novel SRM framework to integrate several key improvements for semanticalization in generation, the associated order, and the sampling strategies. The experimental results are encouraging to some degree. The key idea of this work is to train a spatial reasoning models to jointly denoise multiple variables but with individual noise levels, which is inspired by the previous diffusion forcing work. The authors futher introduce a two-step sampling strategy to overcome the

**Claims And Evidence:**

The key claims of this work are in introducing a SRM (Spatial Reasoning Model) to perform spatial reasoning given human-designed images, following principles such as Sudokus, counting pixels, or counting polygons. It requires advanced visual reasoning to address these tasks reliably. The authors only study the limitations of applying conventional parallel diffusion models and apply a diffusion forcing-like scheme to overcome these challenges.

One fundamental limitation of this paper is the question of why not use the latest VLMs, such as GPT-4o, Gemini-2.0, or other open-source VLMs, to address this symbolic visual reasoning task?

**Essential References Not Discussed:**

The authors are required to provide a discussion of related work on the latest study [1] that attempts to use the latest VLMs to address the constructed visual reasoning tasks.

[1] https://github.com/SakanaAI/Sudoku-Bench

**Experimental Designs Or Analyses:**

The experiment section of this paper is relatively weak and far from convincing, although the spatial reasoning task is a valuable topic. The authors are encouraged to conduct more ablation experiments to analyze why the diffusion forcing scheme and the proposed improvements are important, and how the denoising scheme can understand and address a visual reasoning task like Sudoku.

The current experiments are too superficial and cannot provide deep insights for the community.

**Methods And Evaluation Criteria:**

The major contribution of this work is studying a non-trivial visual reasoning task using generative models such as diffusion or flow matching techniques. A major concern is why not use an LLM or VLM to address such visual reasoning tasks, as Sudokus can be transformed into either a text-only mathematical reasoning task or an image-based math-VQA task. It is confusing for me to understand why the diffusion forcing scheme was chosen to perform denoising on the masked regions.

**Other Comments Or Suggestions:**

No other comments.

**Other Strengths And Weaknesses:**

The major concern of this paper is why the diffusion forcing techniques are used to address a visual reasoning task that existing VLMs might excel at. As mentioned earlier, why not use an LLM or VLM to address such visual reasoning tasks, as Sudokus can be transformed into either a text-only mathematical reasoning task or an image-based math-VQA task? It is confusing for me to understand why the diffusion forcing scheme was chosen to perform denoising on the masked regions.

**Questions For Authors:**

Please refer to the aforementioned weaknesses, and I would like to see how the authors address the major concerns regarding the motivation for using diffusion forcing to tackle the visual reasoning task.

**Relation To Broader Scientific Literature:**

The authors are encouraged to extend the proposed approach to study more challenging visual reasoning tasks at which modern VLMs perform poorly, rather than focusing on naive tasks at which VLMs excel.

**Theoretical Claims:**

The authors provide a thorough theoretical analysis of the Generation Process, Graph-Sequential Sampling, Uniform Sampling, and Recursive Allocation Sampling; however, I have not carefully checked their correctness.

---

> ### Author Rebuttal · Authors · 2025-04-01
>
> We sincerely thank you for your valuable feedback. To address your concerns, we present a point-to-point response in the following:
>
> > Why not use the latest VLMs [...] to address this symbolic visual reasoning task?
>
> Thank you for this important question. Our response is threefold:
>
> 1. The goal of our paper is not to propose the best-performing method for individual visual reasoning tasks from all possible approaches including VLMs. We aim for **evaluating and improving reasoning capabilities of denoising generative models for continuous, spatial domains**. Just like for LLMs, hallucinations are an issue, especially in the case of complex spatial dependencies. Our different strategies can reduce them by a significant amount. We will make sure to resolve any lack of clarity in the final version.
>
> 2. **TLDR: We tested LLMs and VLMs. They do not naively solve Sudoku.**
>    We agree that “Sudokus can be transformed into either a text-only mathematical reasoning task or an image-based math-VQA task.” To evaluate LLMs for this, we conducted an experiment for Sudoku solving with GPT-4o, GPT-4o-mini, and the open-source Phi-4, given full context about the game of Sudoku including its rules, the required output format, and few-shot completion examples (similar to the text-only evaluation of the mentioned Sudoku-Bench). As we noticed deviations from the required format, we resample until obtaining a 9x9 grid of numbers that can be evaluated. Please check out the following figure with quantitative results for 10 samples per number of masked cells: [https://figshare.com/s/b22f205dd69a815d2b89](https://figshare.com/s/b22f205dd69a815d2b89) . While SOTA LLMs (GPT-4o) are able to correctly complete Sudokus with up to 10 missing numbers, their performance quickly deteriorates for more cells to fill. With increasing difficulty, they further start to violate the completion task or in other words cheat by overriding given cells. The following table compares the accuracy of diffusion, our SRM, and LLMs:
>
> | Method | Easy | Medium | Hard |
> | :---- | :---: | :---: | :---: |
> | Diffusion | 0.994 | 0.536 | 0.008 |
> | SRM (Ours) | 0.998 | 0.754 | 0.516 |
> | GPT-4o-mini | 0.205 | 0.000 | 0.000 |
> | GPT-4o | 0.556 | 0.001 | 0.011 |
> | Phi-4 | 0.038 | 0.000 | 0.000 |
>
> Please note that this is not a fair comparison. While we train the diffusion model and SRM for **inpainting of continuous** **visual Sudokus**, the LLM evaluation is a case of **few-shot discrete text completion**. Despite the simpler discrete representation instead of grids of MNIST numbers, LLMs perform poorly in Sudoku completion. Since image-based math-VQA with VLMs can be considered to be an even more difficult task, as it adds correct (implicit) discretization as a step before reasoning, we argue that (visual) Sudoku is not a “naive task at which VLMs excel”. We further support this by showing qualitatively that current systems like ChatGPT (Pro) are not able to perform the full image-to-image task of visual Sudoku (even with few-shot examples): [https://figshare.com/s/dd4d6ad9a281cad7bacf](https://figshare.com/s/dd4d6ad9a281cad7bacf)
>
>
>
> 3. The same goes for the Counting Polygons FFHQ dataset. Please note that the task is not to count polygons and their vertices in input images, but to **generate images** that correctly follow the rules of the data distribution.
>
> > Latest study (Sudoku-Bench) on visual reasoning with VLMs.
>
> Thank you for the pointer to this interesting benchmark. We will add a discussion to the final version. Please note the following:
>
> 1. The benchmark comprises puzzles with different rules that can be leveraged for evaluation of LLMs and VLMs but not for training and testing continuous denoising generative models.
> 2. Just like for our additional experiments with LLMs above, the motivation and experimental setup is different in terms of the representation for reasoning (discrete vs continuous) and in-context learning with given rules (L/VLMs) versus fitting of the correct distribution given training samples only and notably no explicit puzzle rules (SRMs).
> 3. The blog post and GitHub repository were published on the 21st of March 2025, i.e., 4 days before the beginning of the rebuttal phase. Therefore, we hope this will not be considered to be a weakness of our paper.
>
> > “Diffusion Forcing scheme”
>
> Our approach does not simply adopt Diffusion Forcing (DF). We propose a general framework for reasoning over sets of continuous random variables with many possible applications such as spatial domains, but also temporal sequences, which DF is limited to. While the order of sequentialization is fixed to be the temporal order in DF, we propose task-specific (based on a given dependency graph) as well as task-agnostic orders leveraging predicted uncertainty. Moreover, we propose a technique for noise level sampling with larger numbers of variables, whereas the naive approach from DF fails completely in such a setting.

---

### Decision · Program_Chairs · 2025-05-01

**Decision:**

Accept (poster)

**Comment:**

This paper proposes Spatial Reasoning Models (SRMs), a framework leveraging denoising generative models to perform continuous spatial reasoning tasks. It demonstrates that model accuracy improves significantly (e.g., from <1% to >50% on Sudoku) when sequentialization and uncertainty-guided ordering are used. Reviewers appreciated the novelty and careful exploration of denoising strategies but raised concerns about the limited real-world applicability, clarity of theoretical contributions, and modest improvements on more realistic benchmarks. The authors addressed these concerns effectively in the rebuttal, adding new results, clarifying methodology, and reporting comparative results with LLMs/VLMs. AC feels the paper offers meaningful insights into reasoning with generative models, supported by strong empirical improvements. Therefore, an accept decision is recommended.